# Computational basis of hierarchical and counterfactual information processing

Mahdi Ramadan[1,3], Cheng Tang[1,3], Nicholas Watters[1] & Mehrdad Jazayeri ●[1,2] ✉

Humans solve complex multistage decision problems using hierarchical and counterfactual strategies. Here we designed a task that reliably engages these strategies and conducted hypothesis-driven experiments to identify the computational constraints that give rise to them. We found three key constraints: a bottleneck in parallel processing that promotes hierarchical analysis, a compensatory but capacity-limited counterfactual process, and working memory noise that reduces counterfactual fidelity. To test whether these strategies are computationally rational—that is, optimal given such constraints—we trained recurrent neural networks under systematically varied limitations. Only recurrent neural networks subjected to all three constraints reproduced human-like behaviour. Further analysis revealed that hierarchical, counterfactual and postdictive strategies—typically viewed as distinct—lie along a continuum of rational adaptations. These findings suggest that human decision strategies may emerge from a shared set of computational limitations, offering a unifying framework for understanding the flexibility and efficiency of human cognition.

The human brain finds solutions to complex multistage problems that are far more flexible than those learned by artificial systems. Cognitive theories attribute this flexibility to specific algorithms such as hierarchical information processing and counterfactual reasoning[1–7]. Counterfactual reasoning is an important building block of our mental landscape enabling us to imagine alternative accounts of our prior experiences[4,5,8]. A familiar scenario is when we feel the need to revisit our assumptions due to an unexpected turn of events, for example, an unexpected fork in the road while driving, an abrupt emotional reaction from a friend or a twist in the storyline of a book or movie. These situations compel us to review past events and assumptions and look for alternative interpretations that could provide a plausible explanation for our observations. Given the centrality of counterfactuals in cognition, it behooves us to understand the computational underpinnings of this strategy.

Humans commonly rely on counterfactuals when facing decision trees with a hierarchy of if–then scenarios leading to different outcomes[9–13]. However, what is puzzling is that decision trees do not inherently require computing counterfactuals. On the contrary, the optimal inference strategy for solving a decision tree is to rely on the posterior belief over all states within the tree[14–16]. However, in complex problems, computing the posterior is extremely demanding and sometimes intractable[17,18]. Curiously, these are also the conditions in which humans rely on counterfactuals, suggesting that counterfactual reasoning may be a crutch when we cannot compute the posterior. However, several important questions about the computational basis of counterfactual reasoning have remained unanswered. What computational constraints motivate reliance on counterfactuals? Do humans adopt a computationally rational approach to using counterfactuals? Does computing counterfactuals introduce suboptimalities in behaviour? If so, what are they?

We developed a simple and intuitive decision-making task to answer these questions. The task required participants to infer the path of an invisible moving ball within a maze using partial and uncertain cues (Fig. 1a). The task featured two key desiderata needed to investigate the computational characteristics of hierarchical decision-making and counterfactual information processing. First, the maze geometry confronted participants with a two-stage hierarchically organized decision-making problem. Second, parametric control of uncertainty at each decision stage afforded parametric control

[1]Department of Brain and Cognitive Sciences, McGovern Institute for Brain Research, MIT Massachusetts Institute of Technology, Cambridge, MA, USA. [2]Howard Hughes Medical Institute, Cambridge, MA, USA. [3]These authors contributed equally: Mahdi Ramadan, Cheng Tang. ✉e-mail: mjaz@mit.edu

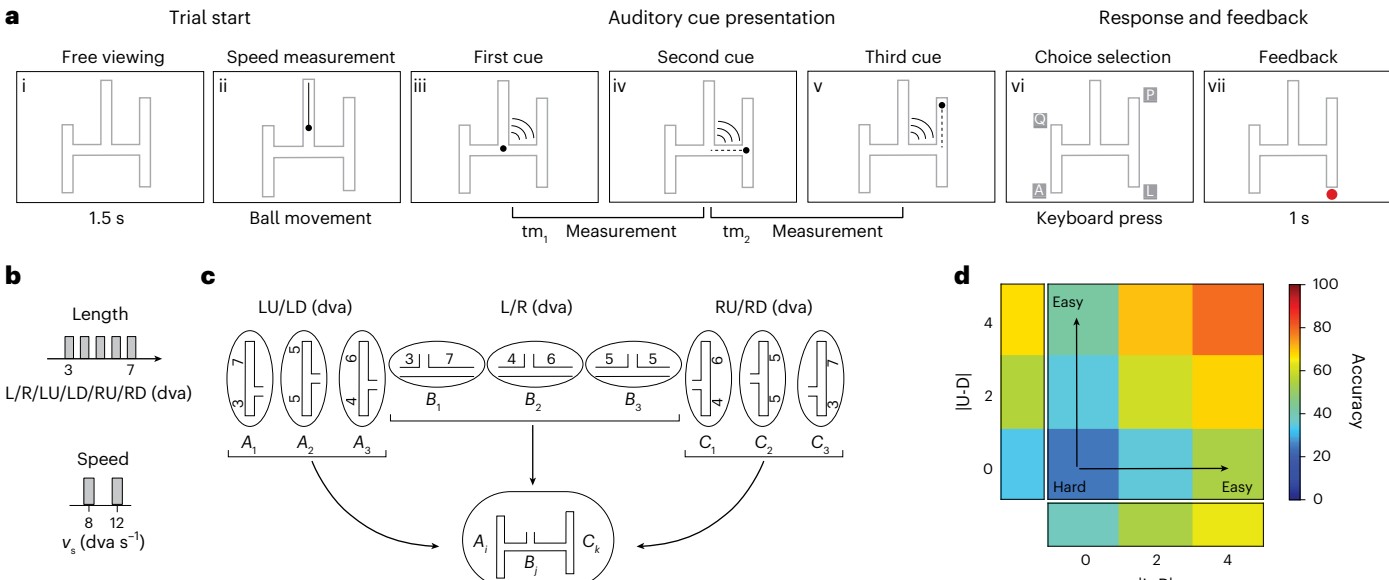

**Fig. 1 | The H-maze task and performance. a**, Task. (i) The participant is presented with an H-shaped maze and four possible choices. (ii) A ball moves visibly towards the maze with constant speed. (iii) An auditory cue signals when the ball reaches the horizontal segment at which time the ball is made invisible and continues to move leftwards or rightwards. (iv) A second auditory cue signals when the ball reaches the vertical segment and the ball changes direction upward or downward (tm₁: the time between the first and second cues). (v) A third auditory cue signalled when the ball reached the endpoint (tm₂: the time between the second and third cues). (vi) The participant presses one of four keyboard buttons to indicate their belief about the final exit point. (vii) The participant receives binary visual feedback only at the location of their choice.

**b**, Top: H-maze arm lengths, denoted by L, R, LU, LD, RU and RD, were 3, 4, 5, 6 and 7 dva. Bottom: the speed of the ball was either 8 or 12 dva s⁻¹, chosen randomly. **c**, Top: we considered three length combinations for each arm pair, either (3, 7), (4, 6) or (5, 5) dva. These pairs are denoted by $(A_1, A_2, A_3)$ for LU/LD, $(B_1, B_2, B_3)$ for L/R and $(C_1, C_2, C_3)$ for RU/RD. Bottom: each H-maze was constructed from connecting one LU/LD pair to one L/R pair to one RU/RD pair as shown in the bottom example in the box. **d**, Performance averaged across participants, showing the percentage of correct responses as a function of absolute length difference between the two horizontal arms (|L − R|) and the two vertical arms associated with the correct horizontal arm (|LU − RD| for left and |RU − RD| for right). The rectangles to the left and bottom show marginal distributions.

over the degree to which counterfactual revisions could improve decisions. Comparing participants' behaviour with inference models implementing various cognitive algorithms, we found that humans use a hierarchical strategy to solve the task sequentially and, when uncertain, revise their decisions by considering counterfactuals. In addition, participants' reaction time profiles and eye movements provided evidence that they relied on hierarchical and counterfactual reasoning to solve the task.

We then used a set of experiments to reverse engineer the computational constraints from which these algorithms derive. Experiment 1 indicated that the hierarchical strategy results from a bottleneck associated with processing parallel streams of evidence. Experiment 2 indicated that compensatory counterfactuals are imperfect owing to working memory limits. Experiment 3 indicated that humans are computationally rational in that the degree to which they rely on counterfactuals depends on the fidelity of their working memory.

Next, we used a modelling approach to test the importance of these computational constraints on adopting a counterfactual strategy. We trained multiple artificial recurrent neural network (RNN) models to perform the same task as humans and subjected each to different subsets of those constraints. Unconstrained models adopted an optimal strategy that deviated substantially from the human participants' behaviour. The addition of constraints altered the model's behaviour. For example, a processing bottleneck constraint shifted the model's responses towards a hierarchical strategy, and working memory limits impact the degree to which the model relied on counterfactuals. Remarkably, the model that most accurately emulated humans' behavioural response patterns was the one that was subjected to all the constraints inferred from human's behaviour. Finally, parametric analysis of the models revealed that distinct cognitive algorithms such as optimal, counterfactual, postdictive and hierarchical may be more

accurately characterized as subdivisions in a continuum that neural systems may adopt depending on the task and computational constraints.

## Results

### The H-maze task

Our task involves inferring the position of a ball travelling in a hierarchical maze based on the timing of auditory cues given when the ball encounters a junction in the maze. On each trial, a ball approaches the horizontal segment of an H-shaped maze from above. Upon reaching the maze, the ball becomes invisible and moves along either the left or the right horizontal arm. Upon reaching the corresponding vertical segment, the ball turns upwards or downwards along one of the vertical arms and stops after reaching the corresponding exit. After the ball stops, participants must report the exit point. Participants must infer the ball's trajectory using partial and ambiguous information provided by three brief auditory clicks. The first click occurs when the ball turns into one of the horizontal arms, the second when it enters one of the vertical arms and the third when it reaches the exit. The time between the first and second clicks ($t_h$) provides information about the horizontal arm the ball enters, and the time between the second and third clicks ($t_v$) provides information about the subsequent vertical arm. Participants must measure these two time intervals and choose the path whose horizontal and vertical arms were compatible with those measurements. In summary, participants have to infer the correct path from the relative timing of the auditory clicks. The H-shaped maze geometry causes a two-level decision tree with uncertainty at each node. We varied the lengths of both horizontal and vertical arms on a trial-by-trial basis to vary the likelihoods at each node of the decision tree and ensure the participants use a flexible dynamic inference strategy (Fig. 1b). Throughout the Article, we will denote the length of the horizontal arms by L (left arm) and R (right arm) and the four vertical

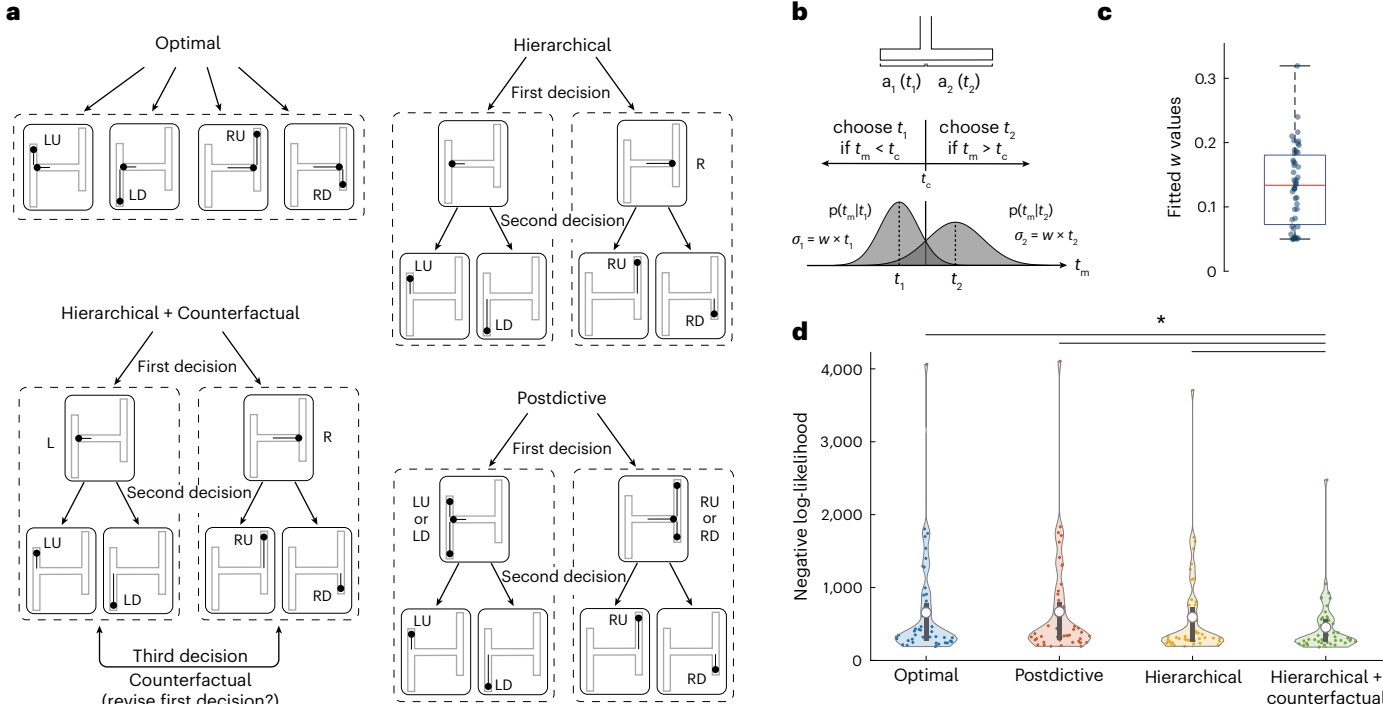

**Fig. 2 | Model-based analysis of behaviour. a**, Plausible computational strategies to solve the H-maze task. Each strategy involves potential decisions (arrows) regarding specific hypotheses (dashed boxes). The symbols inside mazes indicate the arm segments under evaluation. Top left: the optimal model, which decides on the basis of the joint likelihood of the two time intervals. Top right: the hierarchical model, which first chooses left versus right on the basis of the first time interval, and then the corresponding up versus down using the second time interval. Bottom right: the postdictive model, which behaves similarly to the hierarchical model with the key difference that it chooses the horizontal arm postdictively using both time intervals. Bottom left: the counterfactual model. This model behaves similarly to the hierarchical model but can revise its left–right decision if the likelihoods for both vertical arms under consideration are below a threshold. **b**, The T-maze. We used behaviour in the T-maze to measure each participant's timing variability (Weber fraction, $w$). We assumed that participants choose optimally by comparing the two likelihoods—$p(t_m|t_1)$ and $p(t_m|t_2)$—associated with the noisy measurement, $t_m$. The likelihood functions were modelled using a Gaussian distribution with standard deviation scaling with the mean ($\sigma = wt$). **c**, A box plot showing fits to participants' $w$ values ($n = 47$) showing median (line), interquartile range (IQR; box) and minimum and maximum within 1.5 times the IQR (whiskers). **d**, Violin plots of the negative log-likelihood of each model given each participant's responses. The coloured circles represent the mean negative log-likelihood for each participant computed across tenfold cross-validation, while the white circle indicates the mean across all participants. The solid vertical black bar represents the interquartile range (25th and 75th percentiles). The negative log-likelihood of the counterfactual model was significantly lower than that of the other models in capturing participants' choice patterns (left-tailed $t$-test, counterfactual versus optimal: $P = 5.002 \times 10^{-5}$, counterfactual versus postdictive: $P = 2.646 \times 10^{-5}$, counterfactual versus hierarchical: $P < 0.001$, $n = 47$). The outlier with a large negative log-likelihood corresponds to the participant with a large timing variability (outlier in **c** with $w > 0.3$).

arms by LU (left-up), LD (left-down), RU (right-up) and RD (right-down). We may also express arm lengths in units of time reflecting how long it would take for the moving ball to traverse that arm. When doing so, we will use the corresponding labels $t_L$ (left arm), $t_R$ (right arm), $t_{LU}$ (left-up), $t_{LD}$ (left-down), $t_{RU}$ (right-up) and $t_{RD}$ (right-down).

We preregistered our hypotheses and analyses in the Open Science Framework[19] and recruited a large cohort of online participants through the Prolific platform for data collection (Methods). Analysis of behaviour indicated that participants learned the task and were able to use the timing cues to infer the exit point. Average performance across participants improved as a function of the difference between both the horizontal and vertical arms (Fig. 1c). We therefore proceeded with a model-based analysis of behaviour to infer the participant's cognitive strategy.

## Models of different cognitive computational strategies

To quantitatively investigate the strategy participants used for solving the task, we considered a range of models implementing different cognitive strategies with different degrees of optimality. Our first model, which we refer to as the optimal model, chooses the exit point on the basis of a maximum-likelihood strategy; it chooses the path whose composition of horizontal and vertical arms is most consistent with the joint distribution of $t_h$ and $t_v$ (Fig. 2a, top left). Our second model implements a hierarchical strategy by breaking down the problem into two hierarchically organized sequential decisions (Fig. 2a, bottom left). It first uses $t_h$ to choose between the horizontal arms and then uses $t_v$ to choose between the corresponding vertical arms. This strategy is suboptimal because it does not allow the decision about the first arm to benefit from information about the second arm and vice versa. For example, the model may commit to the left horizontal arm on the basis of $t_h$ even if $t_v$ is not consistent with either of the two left vertical arms. Our third model implements a postdictive inference strategy (Fig. 2a, top right). This model functions hierarchically but postpones its first decision until after it has received information about the second arm. More specifically, the model's decision about the horizontal arm is conditioned not only on $t_h$ but also on the degree to which $t_v$ is consistent with the corresponding vertical arms. Note that the postdictive model differs from the optimal model in how the likelihoods of the four vertical arms enter the calculations. The optimal model treats the likelihoods of the four vertical arms separately. The postdictive model, by contrast, relies on the sum of the vertical likelihoods for each side to make its left–right decision. This summation makes the postdictive strategy suboptimal because the sum is blind to the individual likelihoods. For example, imagine a trial with a very high and very low vertical likelihood

on the left and two intermediate likelihoods on the right. The optimal model will take full advantage of these differences. The postdictive model, by comparison, will have less discriminative power because the two sums will be similar. Fourth, we developed a hierarchical model that would additionally consider counterfactuals (Fig. 2a, bottom right). This model starts with the two-stage hierarchical strategy. However, if the likelihood of the vertical arms in the second stage is below a certain threshold, it revises its first decision and chooses between the other two vertical arms. In general, counterfactual reasoning is suboptimal because recalling information from memory is imperfect[20,21]. Note that revisions in counterfactual models differ qualitatively from vacillations that occur during simple decision-making[22,23]. Vacillations arise from fluctuation of evidence, whereas counterfactuals arise from a deliberate process that uses late evidence to revise earlier commitments.

## Humans use hierarchical and counterfactual reasoning

Next, we compare participants' performance against the models across different H-maze geometries. If we assume that models can measure $t_h$ and $t_v$ perfectly, their performance would be trivially at 100%. To make the models more comparable to humans, we assumed that time measurements are noisy. Following the scalar property of timing in humans[24], we modelled noise in timing with a Gaussian distribution whose standard deviation scales with the base interval with a scale factor known as the Weber fraction, $w$. We fit $w$ for each participant on an independent T-maze control task (Fig. 2b, top) in which participants had to discriminate between two horizontal arms associated with two time intervals ($t_1$ and $t_2$). Under the assumption of scalar Gaussian noise (Fig. 2b, middle), we computed the maximum likelihood estimate of $w$ based on the participant's performance in the T-maze (Fig. 2b, bottom). We used the fitted $w$ values for each participant (Fig. 2c, inset) for all subsequent model comparisons. Note that we estimated $w$ for each participant using an independent measure based on the performance in the T-maze task. This approach enabled us to compare participants' behaviour in the H-maze task with cognitive models (optimal, hierarchical and so on) without the need to additionally estimate each participant's timing accuracy, providing a more rigorous approach to model comparison. To compare the behaviour of the models and participants, we computed the cross-validated negative log-likelihood of each model given each participant's responses. We found that the participants' behaviour was most accurately captured by the counterfactual model; that is, the choice pattern of participants' choices was most similar to that of the counterfactual model. Across participants, the counterfactual model had the smallest cross-validated negative log-likelihood compared with the other models (Fig. 2d, $n = 47$, counterfactual versus optimal: $P < 0.001$, counterfactual versus postdictive: $P < 0.001$, counterfactual versus hierarchical: $P < 0.001$, left-tailed $t$-test). Note that, with perfect memory, the counterfactual model produces the same responses as the optimal model. However, empirically, counterfactual reasoning deviates from optimality because of the capacity limitations of working memory needed to recall and process past evidence. Accordingly, our counterfactual model had an additional parameter associated with working memory noise to account for how well participants recall past information. We used tenfold cross-validation to ensure that this additional complexity of the counterfactual model was not due to overfitting.

## Eye movement correlates of decision strategy

To further validate the role of hierarchical and counterfactual strategies in solving the H-maze, we sought to analyse participants' eye movements. Previous studies have shown that the pattern of eye movements during spatial reasoning tasks can reveal behavioural strategy[25–30]. Accordingly, we reasoned that participants may make eye movements indicative of their strategy. Specifically, shifts of the eye towards the left or right early in the trial (for example, after the first time interval) would serve as evidence for a hierarchical strategy, and subsequent revisions of gaze position from one side to the other would serve as evidence for

considering counterfactuals. As it is not possible to monitor eye movements through online experiments, we repeated the experiment with a small cohort of new participants in the laboratory where we could carefully measure their eye movements.

Qualitatively, eye movements showed evidence for both hierarchical and counterfactual processes. Specifically, there were left and right eye movements early in the trial and revisions of gaze position late in the trial (Fig. 3a). Quantitative analysis of eye movements during different epochs of the task provided a rich test bed for several hypotheses. First, the overall frequency of hierarchical saccades was relatively constant across maze geometries suggesting that participants adopted a consistent strategy regardless of maze difficulty (Fig. 3b). Second, the frequency of correct hierarchical saccades decreased for mazes with more similar horizontal arms, suggesting that the horizontal saccades reflected participants' initial left–right decision (Fig. 3b). Third, counterfactual eye movements were most frequent in trials involving difficult first decisions, where computing counterfactuals is most advantageous (Fig. 3c, $n = 4$, easy versus hard first decision conditions: $P < 0.001$; intermediate versus hard: $P = 0.0368$, right-tailed $t$-test).

Finally, grounded in the framework of computational rationality, we used counterfactual saccades to ask whether counterfactuals were used to resolve uncertainty and reduce errors. If so, we would expect the proportion of counterfactual saccades to be higher on trials where the initial hierarchical saccade was incorrect compared with those where it was correct. Consistent with this prediction, eye movement analysis revealed a significantly higher proportion of counterfactual saccades after incorrect initial saccades, compared with those after correct initial saccades (Fig. 3d, $n = 4$, $P = 0.0177$, right-tailed $t$-test). Together, results from eye movements provide complementary evidence that participants relied on hierarchical information processing to solve the H-maze task and used counterfactuals rationally when revisions of the first hierarchical decision were warranted.

One prominent feature of the counterfactual model is the presence of a third decision process associated with deciding whether to consider counterfactuals and, if so, subsequently evaluating the alternative arm segments before making a final choice (Fig. 2a). Because each decision increases processing time, an analysis of reaction times could provide additional evidence as to whether and when participants used a counterfactual strategy. An analysis of the counterfactual model indicated that counterfactuals are relied upon more often when the first decision is more difficult, that is, the difference between the two horizontal arms is small. Accordingly, the hypothesis that participants compute counterfactuals predicts that reaction times would increase systematically with the degree of difficulty of the first decision. Consistent with this prediction, participants' average reaction times increased in a graded fashion with the difficulty of the first decision (Supplementary Fig. 1). This finding provides further evidence that, when uncertain, participants computed counterfactuals.

## Computational constraints on decision-making

Participants' decision strategy raises important questions about the underlying computational constraints. First, why do participants not process the alternatives in parallel? Second, why are the counterfactual decisions suboptimal? Finally, is the participants' reliance on suboptimal counterfactuals rational (that is, do participants know when to rely on counterfactuals)? To address these questions, we conducted three additional large-scale experiments on the Prolific platform to address these questions and elucidate the constraints that underlie the participants' strategy.

**Task variant 1.** We reasoned that hierarchical information processing may be due to an inability to process multiple streams of evidence in parallel, in line with previous findings[18,31–33]. To test this possibility, we developed a variant of the task that could be solved only by parallel processing. Participants were presented with four distinct arms, each

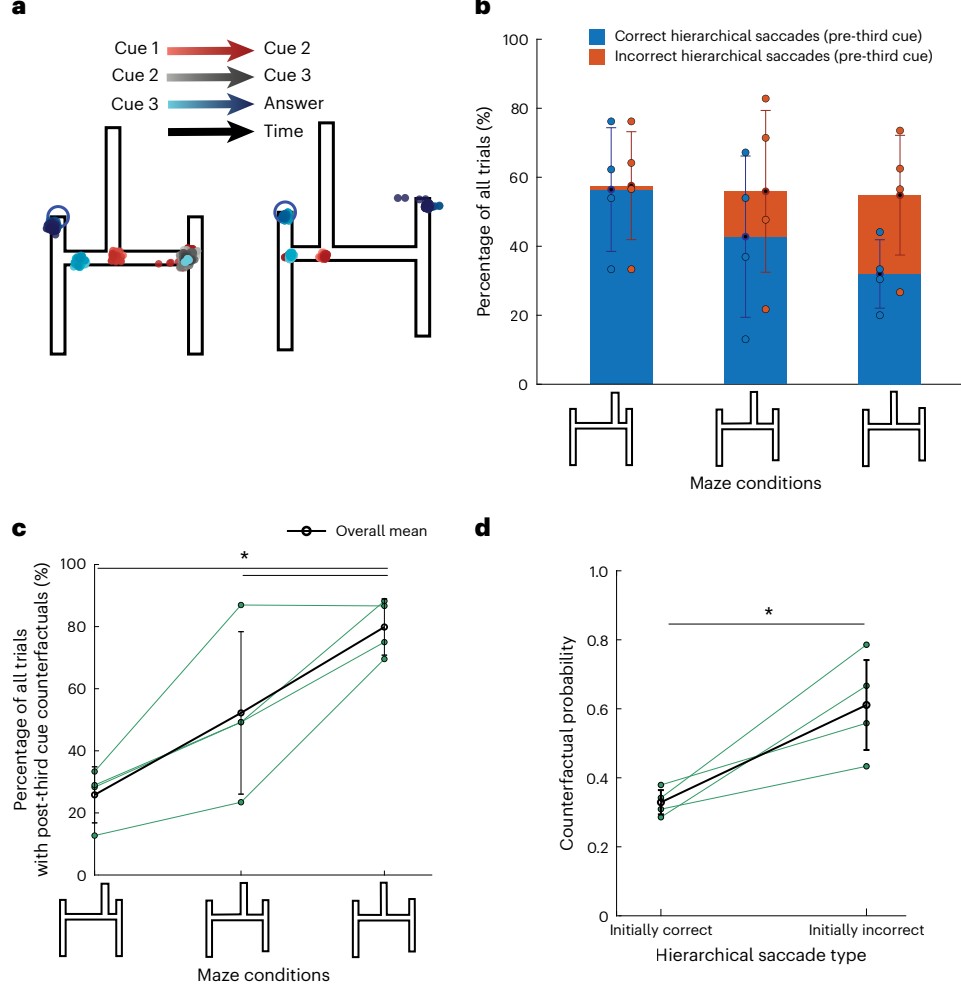

**Fig. 3 | Eye movements. a**, Gaze position density throughout the trial overlaid on the H-maze. Mazes are oriented such that top-left (blue circle) is the correct exit. Gaze positions are colour coded on the basis of the task epoch (red, ball in horizontal segment; grey, ball in vertical segment; blue, ball at endpoint). Bright versus dark represents early versus late in each epoch. The pattern of eye movements are consistent with an early left–right decision (hierarchical) with occasional changes of mind (counterfactual) after the third auditory cue. **b**, The percentage of trials with hierarchical saccades. Stacked bars depict the mean percentage of all trials where participants made hierarchical saccades before the third cue towards either the correct (blue) or incorrect (red) horizontal arm. Results are plotted separately for mazes with different levels of horizontal arm difficulty. Individual participants' data points are overlaid on the bars. The total bar height represents the total percentage of hierarchical saccades (correct and incorrect). The error bars indicate standard deviation across participants ($n = 4$). **c**, The percentage of trials with counterfactual eye movements (black circles) shown for the three horizontal arm difficulty maze conditions. Individual subject data points are overlaid as green circles for each maze condition and are connected with a green line. The error bars indicate the standard deviation across participants. Counterfactual eye movements were most frequent in trials involving difficult first decisions (right-tailed $t$-test; easy versus hard first decision conditions: $P < 0.001$; intermediate versus hard: $P = 0.0368$, $n = 4$). **d**, The probability of counterfactual eye movements based on hierarchical saccade correctness. The black circles show the probability that participants switched their gaze direction (counterfactual behaviour) conditioned on whether the initial hierarchical saccade was made towards the correct or incorrect horizontal arm. Individual subject data points are overlaid as green circles and connected by a green line for the two conditions. The error bars indicate the standard deviation across participants. Participants made significantly more counterfactual saccades when their initially hierarchical saccade was towards the incorrect horizontal arm (right-tailed $t$-test, $P = 0.0177$, $n = 4$). The asterisk shows statistical significance.

serving as a conduit for a separate ball (Fig. 4), mirroring the computational challenge of simultaneously updating posterior probabilities for the four exit points in the original H-maze task to implement the optimal parallel solution. Upon presentation of an auditory cue, the balls entered the four arms moving at constant speed. If a ball reached an end, it reversed direction and continued to move at the same speed. All balls kept moving until the presentation of a second auditory cue that coincided with one randomly chosen ball reaching an arm's end. At that time, participants were offered four choices associated with the four arms and had to report the arm in which the ball was at an exit. Importantly, all task parameters including ball speed, arm lengths and movement times were identical to the H-maze. Moreover, the introduction of direction reversal dissociated the balls' position from time and

forced a strategy in which all balls had to be tracked simultaneously, replicating the computational demands of solving the H-maze task using an optimal decision strategy. Participants' performance on this task was significantly lower than the performance of an ideal observer model solving experiment 1 using fitted $w$ values measured for each participant separately on a control task (Fig. 4, right, $n = 42$, $P < 0.001$, left-tailed $t$-test; see 'T-maze experiment' section in Methods). The relatively low performance associated with parallel processing suggests that participants' choice of a hierarchical strategy is due to a bottleneck associated with processing parallel streams of evidence.

**Task variant 2.** An ideal observer that relies on counterfactuals should, in principle, be able to attain optimal performance because

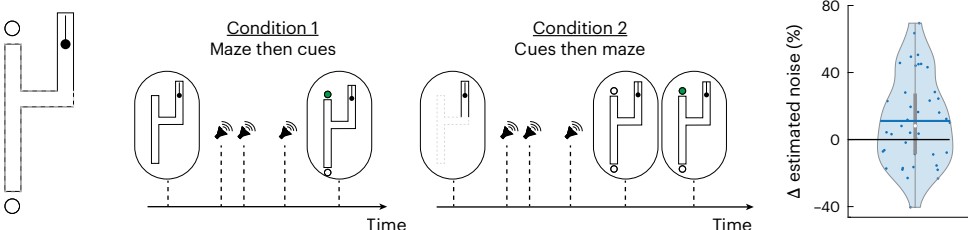

**Fig. 4 | Task variant 1.** Left: a variation in H-maze with independent arms. Four balls moving at a constant and identical speed enter four independent arms simultaneously. Upon entry, an auditory cue is played, and the balls become invisible. The balls continue to move inside their respective arm and reverse direction every time they reach an end. A second auditory cue is played when one of the balls, chosen at random, is at one end of its respective arm (left or right). Subsequently, participants are presented with four options and must choose the arm in which the ball is at its endpoint. Participants receive a binary visual feedback at the location of their choice. Middle: balls move back and forth inside their respective arms until the time of the second auditory cue. In the example shown, the correct answer is the second arm from the bottom. Right: performance for individual participants compared with an ideal observer model solving experiment 1 using fitted $w$ values measured separately for each participant on a control task (horizontal dashed black line, chance-level performance; diagonal dashed black line, unity). The performance of subjects was below the level expected by an ideal observer (left-tailed $t$-test: $P < 0.001$, $n = 42$).

**Fig. 5 | Task variant 2.** Left: a variation of the H-maze with only the left half. Middle: the experiment includes two conditions. In condition 1, the time course of the trial is identical to the original H-maze: stimulus, then the three auditory cues, followed by decision and feedback. In condition 2, participants first hear the three cues and then see the left half of the H-maze, followed by decision and feedback. Right: a violin plot of the percentage change in the estimated noise between condition 2 and condition 1. The estimated noise significantly increases in condition 2 compared with condition 1 (blue circles, individual subjects; coloured solid horizontal line, mean percentage change in estimated noise; white circle, median percentage change in estimated noise; solid vertical grey bar, the 25th and 75th percentiles, two-tailed $t$-test: $P = 0.00741$, $n = 43$).

counterfactuals can be used to evaluate all possibilities sequentially. However, to attain optimal performance, counterfactual processing should be noise-free, or else the performance would drop. Our analysis of participants' behaviour indicated that they relied on counterfactuals, but their performance was suboptimal, which led us to hypothesize that computing counterfactuals may incur additional noise. To test this hypothesis directly, we developed a variant of the task using a simpler T-maze geometry with two conditions (Fig. 5). Condition 1 is analogous to the original H-maze task: the ball enters the maze through a vertical hallway and moves invisibly through the maze while auditory cues indicate the turning points and the time the ball reaches the one of the two exits, and participants are asked to report the exit point. In condition 2, we reversed the order of stimulus presentations such that the T-maze was presented after the presentation of the auditory cues. Importantly, in condition 2, participants must solve the task using counterfactuals, that is, review past evidence from the auditory cues to evaluate alternatives offered by the subsequent presentation of the T-maze. The hypothesis that counterfactual processing is noisy predicts a higher estimated noise in condition 2 compared with condition 1. To test this, we took a model-based approach and computed the maximum likelihood estimate of each subject's noise in both conditions on the basis of the participant's performance (Methods). Results were consistent with this prediction (Fig. 5, right, $n = 43$, $P = 0.00741$, two-tailed $t$-test). Across participants, there was a significant increase in the estimated noise in condition 2 compared with condition 1, revealing the impact of counterfactual processing noise on performance.

**Task variant 3.** Next, we examined the bounded rationality of participants while using counterfactuals. According to the bounded rationality hypothesis, participants should take into account the degrading effect of the counterfactual processing noise and titrate their reliance on counterfactuals accordingly. Specifically, this hypothesis predicts more reliance on counterfactuals for lower counterfactual processing noise and vice versa. To test this hypothesis, we leveraged the performance variance across participants in experiment 2 and asked the same participants to perform a variant of the H-maze task in which they could choose whether and when to rely on counterfactuals (Fig. 6). In this variant, participants were presented with one side of the H-maze (left or right) during the presentation of the auditory cues. Afterwards, participants were given an option to choose between two decision paths. They could either choose one of the two visible exit points as their final answer or ask for the other half of the H-maze to be revealed so that they can counterfactually evaluate the other exit points. Remarkably, on trials where the correct exit point was contained in the hidden half of the H-maze, the proportion of trials in which participants revealed the hidden half of the maze was strongly correlated with their counterfactual processing noise indexed by performance in experiment 2 (Fig. 6, right, $n = 43$, $r^2 = 0.175$, $P = 0.0052$, two-tailed Spearman correlation). By contrast, no such correlation was found on trials in which the original half-maze included the correct exit point (Fig. 6, right, $n = 43$, $r^2 = 1.0 \times 10^{-4}$, $P = 0.962$, two-tailed Spearman correlation). These results reveal the bounded rationality of participants when relying on counterfactuals.

**A neural network model of counterfactual behaviour**
Behavioural responses in H-maze task variants revealed three characteristics of human decision-making: (1) an attentional bottleneck that limits processing multiple streams of information in parallel, (2) counterfactual processing noise that causes suboptimal performance and (3) a judicious use of counterfactuals that takes this processing

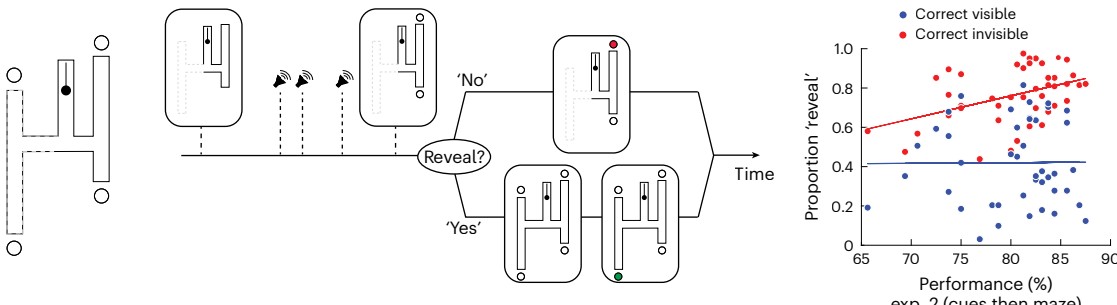

**Fig. 6 | Task variant 3.** Left: a variation on the H-maze in which half of the H-maze is initially visible during the presentation of the auditory cues, and participants may choose to reveal the other half afterwards. Middle: after the presentation of one-half of the H-maze (left or right, chosen randomly) and the three auditory cues, participants may either choose between the two available exit points or ask to have the other half revealed and subsequently choose among the four exit points. Right: the proportion of trials in which participants revealed the hidden side of the maze as a function of performance in condition 2 of experiment (exp.) 2 (cues then maze) across participants. Blue circles are trials where the correct answer was on the revealed half of the H-maze, and red circles are trials where the correct answer was on the unrevealed half of the H-maze. The solid red and blue lines are the best linear fit to the data (two-tailed Spearman correlation, blue: $r^2 = 1.0 \times 10^{-4}$, $P = 0.962$; red: $r^2 = 0.175$, $P = 0.0052$; $n = 43$).

noise into account. However, human experiments alone are not sufficient for evaluating the causal link between these characteristics and the participants' strategies in the H-maze task. To address this shortcoming, we used neural network models, which offer a powerful platform for testing which computational constraints are critical for emulating human-like cognitive strategies[34–37]. Accordingly, we developed multiple task-optimized RNNs and subjected them to different combinations of these constraints to test which ones would generate behavioural response patterns compatible with humans.

Architecturally, the model receives two types of input, has 128 recurrent units and generates two outputs (Fig. 7a). The first type of input is a six-dimensional (6D) vector with values associated with the geometry of the H-maze ($L_{in}$, $R_{in}$, $LU_{in}$, $LD_{in}$, $RU_{in}$ and $RD_{in}$). The second type of input, $I_{time}$, is a two-dimensional (2D) vector specifying the time intervals demarcated by the three auditory cues ($tm_1$ and $tm_2$). The first output specified the model's horizontal choice ($L_{out}$ versus $R_{out}$) and the second output the vertical choice ($U_{out}$ versus $D_{out}$) (Fig. 7a).

Using this architecture, we developed various task-optimized RNNs subject to different sets of constraints. The base model was trained to choose the correct exit without any additional constraints. In the absence of any timing noise, this model could solve the H-maze perfectly without any errors. However, a noise-free model is unrealistic, as it does not account for the scalar variability that influences human timing behaviour. We thus altered $I_{time}$ by adding scalar noise to $tm_1$ and $tm_2$ with Weber fraction $w = 0.15$, which is within the range observed across our participants (Fig. 2c, inset). As expected, introducing timing noise caused the model to make errors. In addition to the base model, we developed various task-optimized RNNs subject to one or more computational constraints inspired by the three experiments on participants' decision-making strategy (Figs. 4–6). The constraints we considered were:

(1) Attentional bottleneck: This constraint modifies training such that the RNN first makes a left–right choice ($L_{out}$ versus $R_{out}$) and then uses this choice to attend to either the left or the right pair of vertical arms. We implement this constraint using a soft attentional gate that biases the input associated with the vertical arms to the left or right side of the H-maze (Methods). In effect, this constraint forces the RNNs to solve the task hierarchically.

(2) Counterfactual processing noise: This constraint modifies training such that the timing information the RNN relies on becomes progressively less reliable for each counterfactual revision. We implement this constraint by adding noise to $tm_1$ and $tm_2$ upon each revision (Methods). This modification is analogous to assuming that the system is subject to counterfactual processing noise.

(3) Rationality: This constraint modifies training such that the RNN learns to choose the exit that is most consistent with its noisy time interval measurements (as opposed to the correct exit). We implement this constraint using a cost function that enforces a maximum-likelihood decision strategy (Methods). This is, in effect, a self-consistency constraint because decisions are made only on the basis of information that is available to the model. Note that the base model violated this rationality constraint because it was optimized with perfect labels that were not available to the learner.

We evaluated the response patterns of all RNN variants by computing the log-likelihood of the RNN's choices relative to those associated with different cognitive strategies (Fig. 7b and Methods). RNNs without the attention bottleneck produced response patterns associated with the optimal strategy (Fig. 7b, columns 1 and 2, $n = 50$, $P < 0.001$, one-tailed $t$-test). This result is expected because RNNs without the attention bottleneck can simultaneously evaluate all possible exits. Incorporating the attention bottleneck caused the RNN to produce response patterns associated with the hierarchical strategy (Fig. 7b, columns 3 and 4, $n = 50$, $P < 0.001$, one-tailed $t$-test). The addition of the rationality constraint enabled the RNN to shift its attention between the two sides and generate responses similar to the optimal model (Fig. 7b, column 5, $n = 50$, $P < 0.001$, one-tailed $t$-test). Finally, adding counterfactual processing noise enabled the RNN to balance the benefit of computing counterfactuals with the degrading effect of the counterfactual noise. The resulting model produced responses that combined the hierarchical and counterfactual strategies and best matched our participants' responses (Fig. 7b, column 6, $n = 50$, $P < 0.001$, one-tailed $t$-test). We refer to this RNN, which was subjected to all constraints, as $RNN_{best}$.

Next, we took a closer look at the $RNN_{best}$ to assess its behaviour more carefully (Fig. 8). We first verified that the overall performance of $RNN_{best}$ was similar to that of human participants and that its responses were sensitive to both horizontal and vertical arm length differences (Fig. 8a). Next, we analysed the 'attention' signal in $RNN_{best}$ within single trials to assess whether the network relied on a counterfactual strategy. A purely hierarchical strategy would predict that $RNN_{best}$ would choose left or right after $tm_1$ and one of the corresponding vertical arms after $tm_2$. In other words, under the hierarchical strategy, $RNN_{best}$ would never switch its attention signal to the alternate horizontal arm after $tm_2$. By contrast, under the counterfactual strategy, $RNN_{best}$ would occasionally switch its left–right attention depending on arm lengths and $tm_2$.

We found that $RNN_{best}$ single-trial behaviour was indeed consistent with the counterfactual strategy. When the difference between

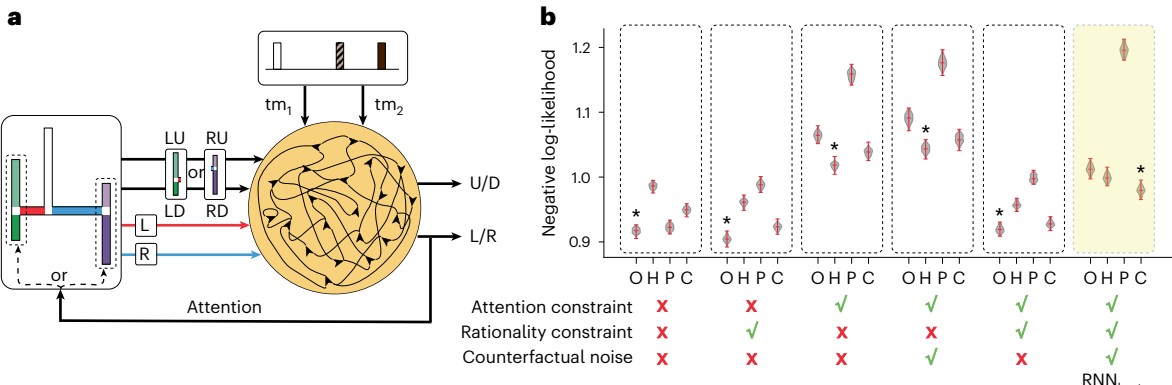

**Fig. 7 | RNN with different constraints implement different decision strategies.**
**a**, RNN architecture. The RNN receives two types of input, a 6D vector specifying the arm lengths of the H-maze ($L_{in}$, $R_{in}$, $LU_{in}$, $LD_{in}$, $RU_{in}$ and $RD_{in}$) and a 2D vector specifying the noisy timing information for interval between the consecutive auditory cues ($tm_1$ and $tm_2$). The RNN produces two outputs. The first output specified the model's horizontal choice ($L_{out}$ versus $R_{out}$), and the second output, the vertical choice ($U_{out}$ versus $D_{out}$). The horizontal output also serves as an attentional bottleneck forcing the RNN to choose between the vertical arms on one side of the maze, which changes the input it receives. **b**, A comparison of the RNN variants to behaviour expected from different cognitive strategies. We trained six task-optimized RNN variants using different combinations of

constraints. The plot shows the negative log-likelihood (NLL) of choices made by each RNN variant conditioned on the four cognitive strategies (O, optimal; H, hierarchical; P, postdictive; C, counterfactual). The legend below the abscissa shows the subsets of constraints (attention bottleneck, rationality and counterfactual noise) that were included (check mark) or excluded (X mark) for each RNN variant. Each violin plot's whiskers mark the maximum, median and minimum of the NLLs obtained from 50 model initializations. Asterisks mark the most likely strategy to each RNN variant ($P < 0.001$, one-tailed $t$-test against the second-lowest strategies, $n = 50$). The yellow band highlights the RNN with all three constraints present, which is the variant that best matched the participants' counterfactual strategy. We refer to this variant as $RNN_{best}$.

horizontal arm lengths was large, $RNN_{best}$ adopted a hierarchical strategy with rare revisions (Fig. 8b, left and middle). In these conditions, the final decision was usually correct when the difference between the corresponding vertical arms was large (Fig. 8b, left), and sometimes wrong when the vertical arms were more similar (Fig. 8b, left). Remarkably, when the difference between the horizontal arm lengths was smaller, $RNN_{best}$ would sometimes revise its first left–right decision (as inferred from the attention signal) depending on $tm_2$ (Fig. 8b, right). This attention-switching behaviour increased systematically for smaller differences between the horizontal arms for which the first decision was more uncertain (Fig. 8c). These behavioural characteristics provide direct evidence that $RNN_{best}$ solves the task using a combination of hierarchical and counterfactual strategies.

Finally, we probed the parametric influence of counterfactual processing noise on the behaviour of $RNN_{best}$. Similar to our experiment 3 in humans (Fig. 6), we predicted that the frequency with which $RNN_{best}$ would rely on counterfactuals would decrease for higher levels of counterfactual processing noise. Results were consistent with this prediction; the proportion of attentional switches dropped systematically under higher levels of counterfactual processing noise. One insight gleaned from this analysis was that the network's inferred strategy was qualitatively different for small and large amounts of counterfactual processing noise ($\sigma_{noise}$). On one end of the spectrum, when noise levels were small ($\sigma_{noise} < 0.2$), the network could rely on counterfactuals without any cost, and thus its behaviour was indistinguishable from the optimal model (Fig. 8d). On the other end of the spectrum, when noise levels made counterfactual processing too costly ($\sigma_{noise} > 1.0$), the network rarely relied on counterfactuals, and thus its behaviour was indistinguishable from the hierarchical model. In other words, computations of $RNN_{best}$ were adaptive and exhibited a graceful transition from optimal to counterfactual to hierarchical, with the counterfactual strategy emerging as the best solution for intermediate levels of noise ($0.2 < \sigma_{noise} < 1.0$).

Together, these results indicate that human participants' strategy is consistent with the predictions of the bounded rationality hypothesis governed by an attentional bottleneck and the magnitude of counterfactual processing noise. Moreover, the continuous transition

between strategies implies that the distinct cognitive algorithms could be mere subdivisions in a strategy continuum under the same objective function.

## Discussion

Humans' cognitive capacity limitations make them unable to find optimal solutions when facing moderately complex problems. Yet, we are quite efficient at finding reasonably good solutions. In cognitive sciences, this ability is viewed through the lens of computationally rational bounded optimality, which posits that humans rely on strategies that are optimal within the bounds of their computational capacity. When facing large decision trees, one of the most common strategies humans adopt is to think through a hierarchy of if–then scenarios and, when needed, consider counterfactuals. While numerous studies have found evidence for these strategies[1–3,38], less is known about the computational constraints that motivate these strategies. Here, we used a moderately complex task for which we could generate precise models to predict behaviour under optimal, bounded-optimal and suboptimal strategies. Comparing model predictions with behaviour, we were able to verify that humans used a combination of hierarchical and counterfactual strategies.

Next, we performed a series of complementary experiments to gain insight into the potential constraints that underlie these strategies. In a first experiment, we verified that parallel processing degrades performance, consistent with previous findings[31–33]. This result is consistent with previous work showing humans' inability to process multiple streams of information simultaneously and the importance of an attentional bottleneck to address this limitation[2,39–43]. In a second experiment, we quantified performance degradation due to counterfactual processing, which is probably due to capacity limitations of memory-based computations[2,20,21,44–49]. In a third experiment, we made the intriguing observations of lower likelihood of counterfactual reasoning for subjects for whom counterfactuals incurred a larger performance cost. This finding suggests that humans' use of counterfactuals is not optimal but computationally rational[17,50]. Together, the results of these experiments provided a clear hypothesis for when humans rely on hierarchical and counterfactual processing: hierarchical processing

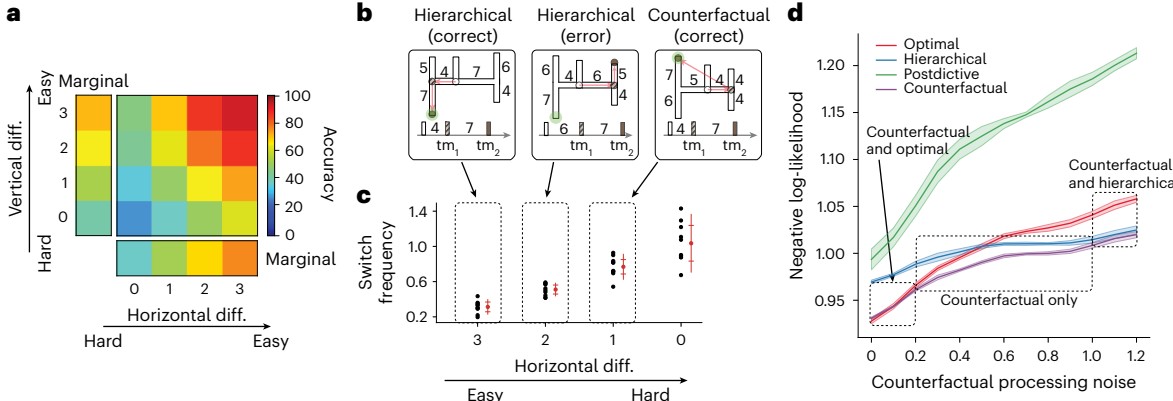

**Fig. 8 | The counterfactual network (RNN_best). a**, Behavioural performance. The accuracy of RNN_best improved systematically as a function of the difference (diff.) between competing horizontal and vertical arms. **b**, Single-trial behaviour of RNN_best for three geometries associated with different arm lengths and different measured time intervals (tm_1 and tm_2). Left: an easy trial where tm_1 and tm_2 match the left bottom exit. Middle: tm_1 and tm_2 are inconsistent with all exits; the RNN first chooses rightwards on the basis of tm_1 and then upwards on the basis of tm_2. Right: tm_1 and tm_2 are inconsistent with all exits; the RNN first chooses rightwards on the basis of tm_1 but, upon seeing tm_2, switches its attention to left and chooses the left-up as its final choice. The two red arrows trace RNN_best decisions after tm_1 and tm_2. The green circle shows the correct answer. See Methods for details.

**c**, The probability of attentional switches as a function of the first choice difficulty (that is, the difference between the horizontal arms is smaller). Results are sampled from ten networks. Whiskers mark the 75th percentile, median and 25th percentile. The three trials in **b** are samples of the three horizontal differences highlighted in **c** (guided by arrows). **d**, The negative log-likelihood (NLL) of observing the choices by RNN_best conditioned on various cognitive strategies as a function of counterfactual processing noise ($\sigma_{noise}$). NLLs are presented as the mean values ± s.e.m. across 50 random model initializations. The dashed blocks show regions with low, intermediate and high $\sigma_{noise}$, which specify the cognitive strategy to which RNN_best is most similar to.

may be used to compensate for performance degradation due to parallel processing, and counterfactual processing may be used judiciously when it can improve performance.

However, human experiments alone do not provide a strong test of these hypotheses because we cannot experimentally manipulate cognitive capacity limitations and computational constraints in participants. To address this shortcoming, we developed task-optimized RNNs that were subjected to different computational constraints and analysed their behavioural patterns to see whether there might be a direct link between computational constraints and cognitive strategies. This modelling effort proved highly fruitful. It provided clear evidence that the models that were subjected to constraints identified in humans found solutions to the H-maze task that were consistent with hierarchical and counterfactual processing, similar to humans. Moreover, the parametric sweeps of the model revealed a direct link between working memory limitation and counterfactual processing. Finally, we found that a single model with a fixed architecture may generate behaviour consistent with the optimal strategy, hierarchical strategy or counterfactual strategy, depending on the computational constraints and noise levels. It is therefore conceivable that strategies that are typically considered as distinct within cognitive sciences may be part of a continuum within neural systems[51]. This finding may open new avenues of exchange between adjacent studies of the brain in systems neuroscience and the mind in cognitive sciences.

## Limitations and future directions

We devised our H-maze task such that it would satisfy certain desiderata. Specifically, we focused on a relatively simple design that involved a hierarchically organized two-stage decision process with experimental control over uncertainty. While this organization lays a foundation for more complex tasks, future work is needed to examine the extent to which our conclusions would generalize to more complex scenarios. Some of the most fruitful directions are (1) studying decision strategies in wider and deeper maze configurations, (2) incorporating other forms of uncertainty such as context switching (for example, covert switches prior statistics such as the speed of the ball) and (3) extending to decision-making tasks in other behavioural domains.

We focused on specific cognitive constraints dictated by the structure of our task, including attentional bottleneck, counterfactual processing noise and rational use of counterfactuals. However, it is likely that the suite of cognitive strategies we rely on including counterfactual reasoning are informed by numerous other constraints that we did not explore. Examples include capacity limitations in processing reward contingencies, integrating sensory evidence across multiple modalities at different timescales, and scenarios where potential outcomes are not known a priori and, thus, have to be inferred through causal reasoning and interventions. Future experiments can build on our computational framework focused on precise quantification of the computational basis of cognitive strategies to explore the larger space of strategies that humans rely on in more naturalistic behavioural settings. These insights may have implications for understanding the human mind in health and disease and for engineering machines with human-like cognitive capacities.

We limited the space of cognitive strategies we considered to optimal, hierarchical, postdictive and counterfactual inference. While these choices cover a set of prominent models in cognitive sciences, they are by no means exhaustive. Furthermore, we did not consider the much larger space of mixed-strategy models, which humans are thought to rely on when facing more complex tasks[52]. For example, participants may choose different strategies upon the initial viewing of the H-maze. One possibility is to adopt a hierarchical strategy when horizontal arms are easily discriminable but choose to ignore the horizontal arms and decide solely on the basis of the vertical arms when they cannot discriminate between the left and right arms. We tested this particular mixed-strategy model and found that it was inferior to the counterfactual model (Supplementary Fig. 2). However, we cannot exclude the possibility that, within the large space of mixed-strategy models, some may capture participants' behaviour better than the counterfactual model. Moreover, we cannot account for idiosyncratic mixed strategies that different participants may use for different mazes. To tackle these limitations, we plan to record neural activity from nonhuman primates trained to perform the H-maze task and use the neural data to make more definitive inferences about the space of cognitive strategies. Neural data could additionally reveal

whether participants rely on the early viewing of the maze to contemplate a plan for which strategy to adaptively use and, thus, may offer a promising opportunity to understand the link between planning and counterfactual reasoning[2,15,38,53–55].

## Methods

We preregistered our hypotheses and analyses for our human psychophysical experiments in the Open Science Framework[19]. All experiments were approved by the Committee on the Use of Humans as Experimental Subjects at the Massachusetts Institute of Technology (protocol number: 1304005676R006, approved on 2 February 2016, renewed on 7 March 2019).

### In-laboratory human participants

Five participants (aged 18–65 years, three males and two females) participated in the in-laboratory eye-tracking studies after giving informed consent. All participants were naive to the purpose of the study, had normal or corrected-to-normal vision and were paid US$12 per hour for their participation. High-fidelity eye-tracking data were successfully calibrated and collected for four participants. For one participant, owing to technical issues, only reaction time data were collected.

In each session, a participant was seated in a dark quiet room and asked to perform the H-maze task for ~60 min. For both tasks, stimuli and behavioural contingencies were controlled by an open-source software (MWorks; https://mworks.github.io/) running on an Apple Macintosh platform. Eye movements were registered by an infrared camera and sampled at 1 kHz (Eyelink 1000, SR Research).

### Extended online participants

In the extended online version of the task, we recruited 150 participants (aged 18–70 years, 87 males and 63 females) on Prolific (https://www.prolific.com/). Participants were informed that the compensation is US$16 per hour. The participants gave consent, and the protocol was approved by the Committee on the Use of Humans as Experimental Subjects at the Massachusetts Institute of Technology.

The participants were divided into 3 groups with 50 participants in each group. Participants in group 1 (27 males, 23 females, 18–55 years old) performed the T-maze (168 trials) and H-maze (540 trials) tasks sequentially in one session (~60 min). Participants in group 2 (28 males, 22 females, 19–70 years old) performed the T-maze (168 trials) and task variant 1 (170 trials) sequentially in one session (~30 min). Participants in group 3 (32 males, 18 females, 22–64 years old) performed the task variant 2 (320 trials) and variant 3 (324 trials) sequentially in one session (~60 min). For task variant 3, the hidden side of the maze was always on the right side, but the correct answer could be on either the left or right side. The tasks were coded in jsPsych (https://www.jspsych.org/latest/) and deployed on the Cognition.run (https://www.cognition.run/) platform.

**Exclusion criteria.** To ensure the quality of online participant data, we exclude subjects whose measured Weber fraction $w$ on the control T-maze experiment or condition 1 of experiment 2 exceeded 0.4.

### Tasks

**Baseline experiment and naming conventions.** The main stimulus in the baseline experiment was a ball moving inside a maze shown from above at a constant speed. The ball entered the maze from the top through a vertical entry hallway, continued moving through the maze at a constant speed and stopped at one of many possible exit points. In the majority of experiments, the maze was shaped like the letter H (hence the name H-maze). In the H-maze experiment, the vertical entry hallway branched into two horizontal arms each of which branched into two vertical arms. We will use these subscripts to refer to the length of specific arms ($a_L$, $a_R$, $a_{LU}$, $a_{LD}$, $a_{RU}$ and $a_{RD}$) as well as the time it would take the ball to move along those arms ($t_L$, $t_R$, $t_{LU}$, $t_{LD}$, $t_{RU}$ and $t_{RD}$). The initial hallways was 7 degree visual angle (dva), and the H-maze arms lengths

were sampled from a discrete uniform distribution with values 3, 4, 5, 6 and 7 dva subject to the constraint that the sum of all arm pairs (L + R, LU + LD, RU + RD) was 10 dva (bottom left). Accordingly, the absolute difference between each arm pair (|L − R|, |LU − LD|, |RU − RD|) could be either 0 (5, 5), 2 (6, 4) or 4 (7, 3) dva (right). The speed of the ball was 8 or 12 dva s⁻¹. With these parameters, the shortest and longest arms lengths (3 and 7 dva) were associated with 0.25 and 0.875 s, respectively. The width of all arms was 0.5 dva.

In the H-maze, the ball (1) started moving downwards along the entry hallway until reaching the horizontal segment, (2) turned left or right and continued at the same speed until reaching one of the vertical segments and (3) turned up or down and continued at the same speed until reaching one exit point where it stopped. Throughout the paper, we use subscripts L (left) and R (right) to refer to the left and right arms of the horizontal segment. Moreover, we use subscripts LU (left-up), LD (left-down), RU (right-up) and RD (right-down) to refer to the four corresponding vertical arms.

**H-maze experiment.** The trial structure in the H-maze experiment was as follows. (1) An H-maze was presented for 1.5 s. Together with the maze, four circles marking the four exit points were presented (colour: white; diameter 1°). (2) A ball visibly travelled through the initial hallway at one of two randomly sampled speeds. (3) Three identical auditory cues were presented (cue duration 80 ms). The first cue was presented when the ball reached the horizontal segment, the second cue when the ball reached the corresponding vertical segment and the third cue when the ball reached the end of the maze. We will refer to the interval between the first and second cues as the first sample interval, denoted by $ts_1$, and the interval between the second and third cues as the second sample interval, denoted by $ts_2$. (4) Participants had to make a keyboard choice (q for LU, a for LD, p for RU and l for RD) to indicate the exit point corresponding to the end position of the ball. The trial aborted if no answer was made within 2,500 ms. (5) The circle corresponding to the participant's choice changed colour to provide feedback (green if correct, purple if incorrect). (6) Trials were separated by an intertrial interval of 1.5 s.

**T-maze experiment.** The T-maze experiment is identical to the H-maze experiment except that there are vertical arms and the exit points are at the end of the two horizontal arms (L and R). Accordingly, participants had to report their binary decision about the exit by choosing one of two circles presented at the end of the two horizontal arms. The two arm lengths ($a_L$ and $a_R$) were sampled from the same distribution as the H-maze. The speed of the ball was also the same as the H-maze task. Therefore, the distribution of the time intervals associated with the two arms ($t_L$ and $t_R$) was also the same as the H-maze task. We performed this experiment to estimate each participant's timing variability parameterized by a Weber fraction, $w$ (see below).

**Task variant 1.** Task variant 1 was designed to force participants to implement the optimal model, where the participant updates their beliefs about all four possible ball trajectories simultaneously. To do so, (1) participants were presented with a variant of H-maze where the four possible ball pathways are flattened into four parallel options, and the timing statistics (for example total time of occluded ball movement) were kept identical between the H-maze task and task variant 1. (2) After a 1.5-s viewing period, four balls initialized in the same location moved visibly towards the maze with identical and constant speed. (3) An auditory cue signalled when the balls reached the horizontal arms at which time the balls were made invisible and continued to move rightwards. If a ball reaches the left or right end of its arm, the ball reverses direction. Crucially, the nonlinearity introduced by the ball direction reversal dissociated the balls' position from time, thus forcing participants to keep track of four ball trajectories separately, and no cue was provided at the time of any ball direction reversal (thus, participants cannot solve this

task hierarchically or sequentially). (4) A second auditory cue signalled when one the balls reached the endpoint of its arm. Crucially, the arm segments and the number of direction reversals were chosen so that the three other balls were not at any arm segment endpoint and were still travelling through their respective arm segments. The endpoint of the ball could be on the right or left end of the arm and after a variable number of direction reversals. (5) Participants had to click one of four circles that appeared next to four possible end points of the ball, and participants had to indicate at which arm end point. (6) Participants received binary visual feedback at the location of choice.

**Task variant 2.** Task variant 2 is similar to the H-maze experiment except that only one half of the maze is shown to the participants ($a_L$, $a_{LU}$, $a_{LD}$ or $a_R$, $a_{RU}$, $a_{RD}$), and thus there are only two exit points similar to the T-maze experiment. The occluded ball travelled through the horizontal arm shown and one of the two vertical arms shown. Accordingly, participants had to report their binary decision about the exit by choosing one of two circles presented at the end of the vertical arms. All arm lengths were sampled from the same distribution as the H-maze. The speed of the ball was also the same as the H-maze task. Therefore, the distribution of the time intervals associated with the two arms ($t_L$ and $t_R$) was also the same as the H-maze task. In condition 1, the participants were shown the half maze and then the auditory temporal cues related to ball movements, just as in the baseline task. In condition 2, however, the order of stimulus presentations was reversed: the half maze was presented after the auditory temporal cues related to ball movements, forcing participants to maintain sensory evidence in working memory and use it afterwards to test different hypotheses about the subsequently shown half maze (that is, counterfactual reasoning). All arm lengths were sampled from the same distribution as the H-maze. The speed of the ball was also the same as the H-maze task.

**Task variant 3.** Task variant 3 is similar to the task variant 2 where initially only one half of the maze is shown to the participants ($a_L$, $a_{LU}$, $a_{LD}$ or $a_R$, $a_{RU}$, $a_{RD}$); however, the true generative process of the ball's movements was identical to the baseline H-maze task (the ball could travel through the revealed horizontal arm and one of the revealed vertical arms, or through the hidden horizontal arm and one of the two hidden vertical arms). The participants were shown the half maze and then the auditory temporal cues related to ball movements. Participants decided the exit point among the two available options in the half maze and subsequently were given the option to reveal the hidden half of the maze, which would allow them to revise their decision and choose between the two new exit ports (that is, revision via counterfactual reasoning). All arm lengths were sampled from the same distribution as the H-maze. The speed of the ball was also the same as the H-maze task.

**Decision model.** Every time the ball reached a bifurcation point, it took one of two alternative arms. We developed a simple decision model based on signal detection theory[56] to characterize how participants discriminated between the two arms using the time between the corresponding two cues. Let us denote the two arm lengths by $a_1$ and $a_2$. Because the ball moves at a constant speed, the arm lengths correspond to the two time intervals, $t_1$ and $t_2$. Depending on which arm the ball takes, the interval between the two cues, denoted $t_s$, will match either $t_1$ or $t_2$. The participant has to use a noisy measure of the actual interval, denoted $t_m$, to decide which of two arms the ball has taken. Consistent with scalar variability of timing[24], we modelled noise in the measurement of time as a sample from a zero-mean Gaussian distribution whose standard deviation scales with the base interval ($t_1$ or $t_2$), with the constant of proportionality $w$. According to this model, the conditional probability of $t_m$ for the two arms can be written as

$$p(t_m|t_1, w_m) = \frac{1}{w_m t_1 \sqrt{2\pi}} e^{-\frac{(t_m - t_1)^2}{2(w_m t_1)^2}}$$

$$p(t_m|t_2, w_m) = \frac{1}{w_m t_2 \sqrt{2\pi}} e^{-\frac{(t_m - t_2)^2}{2(w_m t_2)^2}}.$$

Using signal detection theory[56], the optimal decision is to choose the arm with the higher conditional probability. This is equivalent to choosing the side on which $t_m$ falls relative to a criterion, $t_c$, at the crossing point of the two conditional probability distributions. We refer to this model as the core decision module.

**Estimating each participant's Weber fraction.** We estimated $w$ for each participant by fitting the core decision module to the behaviour in the T-maze task. To accurately estimate $w$, we included $t_c$ as a model parameter to account for idiosyncratic biases. The model also included an extra lapse rate parameter, $\Gamma$, to account for participants' lapses in performance. During lapse trials, the model chooses randomly between the two arms. With these additional parameters, the conditional probability for each arm can be written as

$$p(\text{choose } t_1|t_m) = 0.5\Gamma + (1 - \Gamma)\int_{-\inf}^{t_c} p(t_m|t_1, w_m)\, dt_m$$

$$p(\text{choose } t_2|t_m) = 0.5\Gamma + (1 - \Gamma)\int_{t_c}^{\inf} p(t_m|t_2, w_m)\, dt_m.$$

We computed the maximum likelihood estimate of $w_m$ (as well as the other two parameters) for each participant by fitting this model to behaviour during the T-maze task.

**Optimal model.** This model chooses the most likely exit by comparing the joint conditional distribution of $tm_1$ and $tm_2$ for the four alternatives. The four conditional probabilities can be written as $p(tm_1, tm_2|t_L, t_{LD})$, $p(tm_1, tm_2|t_L, t_{LU})$, $p(tm_1, tm_2|t_R, t_{RU})$ and $p(tm_1, tm_2|t_R, t_{RD})$.

We computed these joint conditional probabilities by the product of corresponding marginals assuming that the two measurements are conditionally independent. Note that this model has no additional parameters other than $w$ that was estimated from the T-maze experiment. Therefore, we did not fit this model to the participants' behaviour; instead, we evaluate its behaviour predictively.

**Hierarchical model.** The hierarchical model solves the H-maze task by making two decisions, first between the horizontal arms and then between the corresponding vertical arms. The first decision chooses between the two horizontal arms by comparing the corresponding conditional probabilities, $p(tm_1|t_L)$ and $p(tm_1|t_R)$. The model chooses the more likely arm and then uses $tm_2$ to choose between the two vertical arms that branch off of the chosen horizontal arm. For example, if the model chooses the left arm in the first stage, it would then compare $p(tm_2|t_{LD})$ and $p(tm_2|t_{LU})$ to choose between the left-up and left-down vertical arms. This model also has no additional parameter other than $w$, which was estimated from the T-maze experiment. Therefore, the behaviour of this model was also evaluated predictively.

**Postdictive model.** The postdictive model is similar to the hierarchical model in that it solves the task by making two hierarchical decisions, first for the horizontal arms and then for the corresponding vertical arms. However, in this model, the first decision for the two horizontal arms is made postdictively by incorporating additional information about the $tm_2$ and the vertical arms. Specifically, the model makes its first decision by comparing $p(tm_1, tm_2|t_L, t_{LD}) + p(tm_1, tm_2|t_L, t_{LU})$ with $p(tm_1, tm_2|t_R, t_{RU}) + p(tm_1, tm_2|t_R, t_{RD})$. Next, it uses $tm_2$ to choose between the two vertical arms that branch off of the chosen horizontal arm, which is identical to the second decision in the hierarchical model. Similar to the optimal and hierarchical model, this model has no additional parameter ($w_m$ is derived from the T-maze experiment).

**Counterfactual model.** The counterfactual model is identical to the hierarchical model but has the flexibility to revise its decisions when a certain measure of expected accuracy ($X$) is lower than a certain threshold ($\theta$) that was fit as a free parameter.

**Noise in the counterfactual model.** To compute the likelihood of a counterfactual possibility, participants must perform an offline mental computation that involves inferring the values of $tm_1$ and $tm_2$ from memory to test a new hypothesis (the likelihood that the exit is down the other horizontal arm). As the process of inferring time intervals from memory is noisy, we formulated the counterfactual model such that the recalled values of $tm_1$ and $tm_2$ (values used during the revision process) were participant to additional noise. Similar to the noisy optimal, hierarchical and postdictive models, the noise in the counterfactual model was sampled from a Gaussian distribution with mean $\beta$ and standard deviation that is proportional to the measured interval ($tm_1$ or $tm_2$) with a fixed constant of proportionality $\alpha$ (in accordance with scalar variability of time). The parameters $\alpha$ and $\beta$ were fitted using maximum likelihood estimation.

**Mixed-strategy model.** This model assumes that participants adapt their decision-making strategy on the basis of the difficulty of the initial decision, quantified by the log-likelihood ratio (evidence). For trials where the initial decision between horizontal arms is relatively easy (|log-likelihood ratio| >threshold), the model solves the task using a hierarchical strategy (see above). For trials where the initial decision is difficult (|log-likelihood ratio| <threshold), the model does not commit to a horizontal arm. Instead, it relies exclusively on the second measurement $tm_2$ to evaluate the four exit points. In this uncommitted case, the model chooses the most likely exit by comparing the four conditional probabilities $p(tm_2|t_{LD})$, $p(tm_2|t_{LU})$, $p(tm_2|t_{RU})$ and $p(tm_2|t_{RD})$. Unlike the counterfactual model, the mixed-strategy model avoids revising an initial decision by withholding commitment under uncertainty. The model introduces a single free parameter, the evidence threshold $\theta$, which determines whether the model commits to an initial decision or remains uncommitted. This parameter was fit and cross-validated to participants' behaviour.

**Estimated noise.** In our preregistration, we proposed a model-free analysis of experiment 2 data based on subject performance for different arm length differences. However, this analysis confounds performance differences owing to stimulus difficulty (arm length difference) with the timing sensitivity of subjects. Here, we conduct an improved model-based analysis of experiment 2 data where we fit a performance-based psychometric curve over all arm length differences for each subject. Let us denote the two arm lengths by $t_1$ and $t_2$, and the difference in measured time between $t_1$ and $t_2$ as $t_d$. To characterize how participants discriminated between the two arms using the time between the corresponding two cues, we fit a cumulative Gaussian model to compute the probability of a correct choice given $t_1$ and $t_2$ as follows:

$$p\left(\text{correct}|t_1, t_2, \sigma\right) = \int_0^\infty \frac{1}{\sigma\sqrt{2\pi}} e^{-\frac{(t_d - |t_1 - t_2|)^2}{2\sigma^2}} \, dt_d.$$

Once we compute the maximum likelihood estimate of each subject's noise $\sigma$ for each condition, we calculate the percentage change in the subjects' estimated noise $\sigma$ between condition 1 and condition 2 as

$$100 \times \frac{\sigma_{\text{condition2}} - \sigma_{\text{condition1}}}{\sigma_{\text{condition1}}}.$$

The hypothesis that subjects' estimated noise increases on condition 2 relative to condition 1 predicts a significantly positive percentage change. The null hypothesis predicts no such significant change.

**Eye-tracking analysis.** Raw eye movement data were preprocessed to remove high-frequency artefacts and offset based on calibration, ensuring accurate alignment of gaze positions. Hierarchical saccades were identified if the $x$-axis gaze positions extended beyond 2 dva from the fixation point before the presentation of the third cue and remained within 10 dva from the fixation point in both the $x$ and $y$ axes, defining the region of interest on the screen. Switching behaviour was defined as a reversal in gaze direction after the presentation of the third cue, representing a potential reconsideration of the initial decision. A switch was detected when the gaze shifted from one side of the maze to the other, adhering to the spatial bounds used for hierarchical saccades.

**The RNN model.** To construct the trials for the RNN, we sample the length of each of the six maze segments [L, R, LU, LD, RU, RD] from the set of lengths {4, 5, 6, 7} uniformly. Thus there are $4^6 = 4{,}096$ possible maze geometries. We randomly split the geometries into 3,072 (75%) for model training and 1,024 (25%) for model testing. When RNN performs the task, we apply noise to the interclick intervals. We convert each ground truth interclick interval $ts_i$ to a perceived interclick interval $tm_i = ts_i(1 + \sigma_{noise}\epsilon)$ where $\epsilon \sim N(0,1)$ ('distributed as') with $\sigma_{noise} = 0.15$. This noise is independently sampled for each of the two interclick intervals per stimulus. For each maze geometry, we consider each of the four possible ball paths and for each of the conditions sample 10 perceived interclick interval pairs ($tm_1$, $tm_2$). Thus, we generate 40 trials for each maze geometry in both the training and testing sets.

To train RNNs on the H-maze task, we discretize time into 32 timesteps per trial. We let the ball begin 5 length units above the horizontal maze arms and travel at 1 unit per timestep, so on every trial the ball reaches the first T-junction at timestep 6. For visual input, we provide models with scalar values of the arm lengths [L, R, LU, LD, RU, RD]. For auditory input, although human subjects received discrete auditory clicks that demarcated time intervals, previous studies have shown that time intervals were represented internally as ramping signals between the discrete clicks in the brain and RNNs[57,58]. To facilitate RNN training, here we directly use the corresponding ramping signal as the time interval input to the RNN. We construct a horizontal and vertical auditory timing variable [$a_1$, $a_2$]. We let $a_1$ be 0 until the ball reaches the first T-junction, then ramp linearly from 0 to the first perceived interclick interval $tm_1$ in int($tm_1$) (discretize to nearest integer) timesteps and remain constant thereafter. Similarly, we let $a_2$ be 0 until $a_1$ reaches $tm_1$, upon which $a_2$ ramps linearly to the second perceived interclick interval $tm_2$ and in int($tm_2$) timesteps then remains constant. The RNN provides two outputs, a horizontal choice and a vertical choice ([$h$, $v$]) that jointly specify the chosen exit.

Our counterfactual RNN model consists of a recurrent state $\mathbf{r}$ of $N = 128$ hidden units, updated each timestep according to

$$\mathbf{r}_{t+1} = \tanh(\{W_{\text{rec}} \, \mathbf{r}_t + W_{\text{in}}\mathbf{I}_{t+1}), \tag{1}$$

where $W_{\text{rec}}$ is an $N \times N$ matrix of trainable recurrent weights, $W_{\text{in}}$ is an $N \times 8$ matrix of trainable input weights and $\mathbf{I}$ is an eight-dimensional input vector [$a_{1t}, a_{2t}$, L, R, LU, LD, RU, RD] at each timestep.

The model produces two choices (horizontal and vertical) at each timestep representing the vertical and horizontal choices

$$\mathbf{h}_t, \mathbf{v}_t = \sigma(W_{\text{out}} \cdot \mathbf{r}_t), \tag{2}$$

where $W_{\text{out}}$ is an $N \times 2$ matrix of learnable weights and $\sigma$ is the sigmoid function $\sigma(x) = 1/(1 + e^{-x})$. We use a threshold of 0.5 to binarize the choice to evaluate behavioural outcomes.

For the basic version of the RNN, we use an external-supervising signal to train the RNN. For each trial, we minimize the MSE between the RNN output [$h_t$, $v_t$] and the ground-truth output. The loss is summed over the timesteps after $tm_2$ finishes ramping.

**Attention constraint.** To add the attentional bottleneck to the RNN, we change the original eight-dimensional input $[a_{1t}, a_{2t}, \text{L}, \text{R}, \text{LU}, \text{LD}, \text{RU}, \text{RD}]$ into a six-dimensional input $[a_{1t}, a_{2t}, \text{L}, \text{R}, U_t, D_t]$, where $U_t$ and $D_t$ are the lengths of vertical arms gated by an attention module. Mechanistically, the attention module produces outputs $U_t$ and $D_t$ with the horizontal choice $h_t$ at each timestep $t$ as follows:

$$U_t = h_t\text{LU} + (1 - h_t)\text{RU}$$
$$D_t = h_t\text{LD} + (1 - h_t)\text{RD}.$$

Accordingly, the input weight matrix becomes $N \times 6$ with the attention constraint.

**Rationality constraint.** Instead of supervised learning, we train the RNN with the objective of maximizing the likelihood of the chosen exit given the noisy timing information, which is the rational choice under uncertainty. Given a maze geometry and perceived interclick intervals $\text{tm}_1$ and $\text{tm}_2$, a rational policy on the H-maze task chooses an exit point corresponding to a path in the maze that maximizes the joint log-likelihood $\log(p(\text{tm}_1|\text{ts}_1)) + \log(p(\text{tm}_2|\text{ts}_2))$. Because $\text{tm}_i$ is sampled from a Gaussian distribution centred at the ground truth $\text{ts}_i$, maximizing the log-likelihood is equivalent to minimizing the loss function

$$\mathscr{L} = \left(\frac{\text{tm}_1 - \text{ts}_1}{\text{ts}_1}\right)^2 + \left(\frac{\text{tm}_2 - \text{ts}_2}{\text{ts}_2}\right)^2.$$

Considering the output logits $[\mathbf{h}_t, \mathbf{v}_t]$ as the model's choice of which side of the maze the ball exits (left/right and up/down), the loss function becomes

$$\mathscr{L}(t) = h_t\left(\frac{\text{tm}_1 - \text{L}}{\text{L}}\right)^2 + (1 - h_t)\left(\frac{\text{tm}_1 - \text{R}}{\text{R}}\right)^2$$
$$+ v_t\left(\frac{\text{tm}_2 - U_t}{U_t}\right)^2 + (1 - v_t)\left(\frac{\text{tm}_2 - D_t}{D_t}\right)^2. \tag{3}$$

On the right-hand side of the equation, the first two terms are the loss coming from the horizontal arm choice $h_t$ and the last two terms are the loss coming from the vertical arm choice $v_t$ given the horizontal arm choice. When training the models, this loss function is summed over the timesteps after $\text{tm}_2$ finishes ramping.

**Counterfactual noise constraint.** To discourage the model from switching its attention from side to side every timestep, we impose an implicit cost for switching: after the second auditory click (namely after $a_2$ finishes ramping), each time h crosses 0.5 we add noise to the perceived interclick intervals $[\text{tm}_1, \text{tm}_2]$. Specifically, for each timestep $t$ where $h$ crosses 0.5, we update tm as

$$\text{tm}_i \leftarrow \text{tm}_i(1 + \sigma_{\text{noise}}\epsilon), \epsilon \sim N(0, 1), i = 1, 2$$

and use this noised tm for all subsequent timesteps in the trial. Here, $\sigma_{\text{noise}}$ is a hyperparameter controlling the magnitude of this switching penalty. We set $\sigma_{\text{noise}} = 0.5$ for the counterfactual noise constraint and $\sigma_{\text{noise}} = 0$ without this constraint.

**RNN variants and model comparison.** There are in total eight possible RNN variants with three constraints that are binary (attention, rationality and counterfactual noise). However, we ruled out two illogical variants: because counterfactual noise is triggered by the reverse of attention, it does not make sense to have counterfactual noise without the attention constraint. The counterfactual strategy used $\sigma_{\text{noise}} = 0.5$ for the counterfactual noise, and the switching threshold was fitted by maximum likelihood estimation with twofold cross-validation.

**RNN saccade.** Although the RNN generates continuous outputs $[h_t, v_t]$ throughout the trial, we plot the analogue of a saccade at two discrete timepoints for clarity (the timesteps when $\text{tm}_1$ and $\text{tm}_2$ finish ramping).

For the first saccade (right after $\text{tm}_1$), we consider only $h_t$ and ignore $v_t$ because the RNN cannot make a vertical decision without $\text{tm}_2$. For the second saccade (right after $\text{tm}_2$), the saccade position reflects both $h_t$ and $v_t$.

## Reporting summary

Further information on research design is available in the Nature Portfolio Reporting Summary linked to this article.

## Data availability

The data collected from in-laboratory and online experiments are available via GitHub at https://github.com/jazlab/MR_CT_NW_MJ_2024.

## Code availability

The code used for modelling and to generate the associated figures is available via GitHub at https://github.com/jazlab/MR_CT_NW_MJ_2024.

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

## Acknowledgements

M.R. is supported by Lisa K. Yang ICoN fellowship. C.T. is supported by Friends of the McGovern Institute Student Fellowship. N.W. is supported by the National Science Foundation Graduate Research Fellowship Program. M.J. is supported by the Simons Foundation, HHMI and the McGovern Institute. The funders had no role in the study design, data collection and analysis, decision to publish or preparation of the manuscript.

## Author contributions

M.R. and M.J. conceived the study. M.R. collected the human psychophysics data and performed all the analyses. C.T., N.W., M.R. and M.J. conceived the RNN architecture. C.T. and N.W. implemented the RNN code. C.T. collected the online experiment data. C.T. performed all RNN analyses. M.R., C.T., N.W. and M.J. wrote the paper. M.J. supervised the project.

## Competing interests

The authors declare no competing interests.

## Additional information

**Supplementary information** The online version
contains supplementary material available at

**Correspondence and requests for materials** should be addressed to
Mehrdad Jazayeri.

**Peer review information** *Nature Human Behaviour* thanks Ariel
Zylberberg and the other, anonymous, reviewer(s) for their
contribution to the peer review of this work. Peer reviewer reports are
available.

# Reporting Summary

## Statistics

For all statistical analyses, confirm that the following items are present in the figure legend, table legend, main text, or Methods section.

| n/a | Confirmed | |
|---|---|---|
| ☐ | ☒ | The exact sample size (*n*) for each experimental group/condition, given as a discrete number and unit of measurement |
| ☐ | ☒ | A statement on whether measurements were taken from distinct samples or whether the same sample was measured repeatedly |
| ☐ | ☒ | The statistical test(s) used AND whether they are one- or two-sided<br>*Only common tests should be described solely by name; describe more complex techniques in the Methods section.* |
| ☐ | ☒ | A description of all covariates tested |
| ☐ | ☒ | A description of any assumptions or corrections, such as tests of normality and adjustment for multiple comparisons |
| ☐ | ☒ | A full description of the statistical parameters including central tendency (e.g. means) or other basic estimates (e.g. regression coefficient) AND variation (e.g. standard deviation) or associated estimates of uncertainty (e.g. confidence intervals) |
| ☐ | ☒ | For null hypothesis testing, the test statistic (e.g. *F*, *t*, *r*) with confidence intervals, effect sizes, degrees of freedom and *P* value noted<br>*Give P values as exact values whenever suitable.* |
| ☐ | ☒ | For Bayesian analysis, information on the choice of priors and Markov chain Monte Carlo settings |
| ☐ | ☒ | For hierarchical and complex designs, identification of the appropriate level for tests and full reporting of outcomes |
| ☒ | ☐ | Estimates of effect sizes (e.g. Cohen's *d*, Pearson's *r*), indicating how they were calculated |

*Our web collection on statistics for biologists contains articles on many of the points above.*

## Software and code

Policy information about availability of computer code

| Data collection | Eyelink 1000, MWorks 0.8, Cognition.run, Prolific, jsPsych |
|---|---|
| Data analysis | Matlab R2022a, Python 3.9 |

For manuscripts utilizing custom algorithms or software that are central to the research but not yet described in published literature, software must be made available to editors and reviewers. We strongly encourage code deposition in a community repository (e.g. GitHub). See the Nature Portfolio guidelines for submitting code & software for further information.

## Data

Policy information about availability of data

All manuscripts must include a data availability statement. This statement should provide the following information, where applicable:

- Accession codes, unique identifiers, or web links for publicly available datasets
- A description of any restrictions on data availability
- For clinical datasets or third party data, please ensure that the statement adheres to our policy

https://github.com/jazlab/MR_CT_NW_MJ_2024/tree/master

# Research involving human participants, their data, or biological material

Policy information about studies with [human participants or human data](). See also policy information about [sex, gender (identity/presentation), and sexual orientation]() and [race, ethnicity and racism]().

| | |
|---|---|
| Reporting on sex and gender | 93 males and 68 females. The sex is randomly sampled in the pool of participants available on Prolific. We have no hypothesis on sex-based difference in the task and have no such analysis in the study. |
| Reporting on race, ethnicity, or other socially relevant groupings | We have no socially relevant categorization variables in the study. |
| Population characteristics | See above |
| Recruitment | We have no filter on participants other than 'fluent English reading'. We published 150 openings of experiment opportunities on Prolific, and qualified participants on Prolific voluntarily participated.<br>The experiments do not require specific skills or backgrounds, and we don't expect any self-selection biases or others. |
| Ethics oversight | Committee on the Use of Humans as Experimental Subjects at the Massachusetts Institute of Technology |

Note that full information on the approval of the study protocol must also be provided in the manuscript.

# Field-specific reporting

Please select the one below that is the best fit for your research. If you are not sure, read the appropriate sections before making your selection.

☐ Life sciences   ☒ Behavioural & social sciences   ☐ Ecological, evolutionary & environmental sciences

For a reference copy of the document with all sections, see [nature.com/documents/nr-reporting-summary-flat.pdf]()

# Behavioural & social sciences study design

All studies must disclose on these points even when the disclosure is negative.

| | |
|---|---|
| Study description | mixed-methods case study on human decision-making strategy for hierarchical problems |
| Research sample | Randomly sampled participants on Prolific. 93 males and 68 females, 18-70 years old. We have no hypothesis on the demographic influence in this study.<br>The sample size of the pre-registered online experiment was chosen empirically to be 10 times the on-site experiment sample size for reliable reproduction. The samples are representative without identifiable biases. |
| Sampling strategy | Random sample. We first peformed the experiment with n=15 on-site participants. Then pre-registered and verified the results on n=150 online participants.<br>We didn't calculate the sample size through statistical methods. The sample size of the pre-registered online experiment was chosen empirically to be 10 times the on-site experiment sample size for reliable reproduction. |
| Data collection | For on-site experiment, stimuli and behavioral contingencies were controlled by an open-source software (MWorks; mworks-project.org/) running on an Apple Macintosh platform. Eye tracking data were collected by Eyelink device. No researcher was present besides the participants and the researcher was blind to the experiment during data collection.<br>For online experiment, the tasks were coded in jsPsych (www.jspsych.org) and deployed on the Cognition.run (www.cognition.run) platform. We only collected the keyboard response from online participants. |
| Timing | On-site experiment: HMaze data was collected 10/20/2018 to 11/02/2018. The variants were collected 9/16/2019 to 10/18/2019.<br>Online experiment: created and finished on 28 Feb 2024 |
| Data exclusions | To ensure the quality of online participant data, we exclude subjects whose measured Weber fraction w on the control T-maze experiment or condition 1 of experiment 2 exceeds 0.4. |
| Non-participation | 17 participant returned their response due to technical problems, 1 participant was dropped out due to time-out |
| Randomization | not allocated into groups |

# Reporting for specific materials, systems and methods

We require information from authors about some types of materials, experimental systems and methods used in many studies. Here, indicate whether each material, system or method listed is relevant to your study. If you are not sure if a list item applies to your research, read the appropriate section before selecting a response.

## Materials & experimental systems

| n/a | Involved in the study |
|-----|----------------------|
| ☒ ☐ | Antibodies |
| ☒ ☐ | Eukaryotic cell lines |
| ☒ ☐ | Palaeontology and archaeology |
| ☒ ☐ | Animals and other organisms |
| ☒ ☐ | Clinical data |
| ☒ ☐ | Dual use research of concern |
| ☒ ☐ | Plants |

## Methods

| n/a | Involved in the study |
|-----|----------------------|
| ☒ ☐ | ChIP-seq |
| ☒ ☐ | Flow cytometry |
| ☒ ☐ | MRI-based neuroimaging |

## Plants

| | |
|---|---|
| Seed stocks | *Report on the source of all seed stocks or other plant material used. If applicable, state the seed stock centre and catalogue number. If plant specimens were collected from the field, describe the collection location, date and sampling procedures.* |
| Novel plant genotypes | *Describe the methods by which all novel plant genotypes were produced. This includes those generated by transgenic approaches, gene editing, chemical/radiation-based mutagenesis and hybridization. For transgenic lines, describe the transformation method, the number of independent lines analyzed and the generation upon which experiments were performed. For gene-edited lines, describe the editor used, the endogenous sequence targeted for editing, the targeting guide RNA sequence (if applicable) and how the editor was applied.* |
| Authentication | *Describe any authentication procedures for each seed stock used or novel genotype generated. Describe any experiments used to assess the effect of a mutation and, where applicable, how potential secondary effects (e.g. second site T-DNA insertions, mosiacism, off-target gene editing) were examined.* |

