## [Peer Review File · Nature Human Behaviour]

Computational basis of hierarchical and counterfactual information processing

Corresponding Author: Professor Mehrdad Jazayeri

A version of this paper was originally rejected for publication by Nature Human Behaviour, however that decision was reconsidered after appeal by the authors.

Version 0:

Decision Letter:

Dear Professor Jazayeri,

Thank you for submitting your manuscript entitled, "Computational basis of hierarchical and counterfactual information processing", and for your patience while awaiting an initial decision.

As you may know, we decline a substantial proportion of manuscripts without sending them to referees based on an editorial evaluation of the extent to which the work meets editorial criteria for suitability of publication in Nature Human Behaviour (please see our [Editorial](https://www.nature.com/articles/s41562-019-0778-0)).

After careful consideration, I regret that we cannot offer to publish this work in Nature Human Behaviour. We found the topic of this research to be interesting, and we also appreciated the novel task. However, we are unable to proceed with this manuscript as we feel that the sample size ($n=15$ total across all experiments) is very low for a study of this nature. We would be interested in reconsidering an expanded version of this work but only if the sample size could be increased by at least an order of magnitude. Please contact me if you have any questions about this possibility.

I am sorry that we cannot respond more positively on this occasion, and hope that the negative outcome in this instance will not deter you from submitting future work to Nature Human Behaviour.

Sincerely,

[Redacted]

[Redacted]

[Redacted]

Nature Human Behaviour

Version 1:

Decision Letter:

17th October 2024

Dear Professor Jazayeri,

Thank you once again for your manuscript, entitled "Computational basis of hierarchical and counterfactual information processing", and for your patience during the peer review process.

Your Article has now been evaluated by 3 referees. You will see from their comments copied below that, although they find your work of considerable potential interest, they have raised quite substantial concerns. In light of these comments, we cannot accept the manuscript for publication, but would be interested in considering a revised version if you are willing and able to fully address reviewer and editorial concerns.

We hope you will find the referees' comments useful as you decide how to proceed. If you wish to submit a substantially revised manuscript, please bear in mind that we will be reluctant to approach the referees again in the absence of major revisions. We are committed to providing a fair and constructive peer-review process. Do not hesitate to contact us if there are specific requests from the reviewers that you believe are technically impossible or unlikely to yield a meaningful outcome.

In particular, it will be important to address the shared concerns of Reviewer #1 and #3 regarding whether the H-maze task is truly a counterfactual reasoning task. Additional data collection may be required.

If you wish to submit a suitably revised manuscript, we would hope to receive it within 4 months. I would be grateful if you could contact us as soon as possible if you foresee difficulties with meeting this target resubmission date.

* Include a "Response to the editors and reviewers" document detailing, point-by-point, how you addressed each editor and referee comment. If no action was taken to address a point, you must provide a compelling argument. When formatting this document, please respond to each reviewer comment individually, including the full text of the reviewer comment verbatim followed by your response to the individual point. This response will be used by the editors to evaluate your revision and sent back to the reviewers along with the revised manuscript.

* Highlight all changes made to your manuscript or provide us with a version that tracks changes.

* Please ensure that the code used in the study is available to be shared with reviewers in the next round.

Link Redacted

Thank you for the opportunity to review your work. Please do not hesitate to contact me if you have any questions or would like to discuss the required revisions further.

Sincerely,

[Redacted]

[Redacted]

[Redacted]

Nature Human Behaviour

REVIEWER COMMENTS:

Reviewer #1 (Remarks to the Author):

This paper examines how humans approach complex, multi-step decision problems. The authors developed an H-shaped maze task in which participants infer the trajectory of an invisible ball using partial and uncertain cues. By integrating human psychophysics with behaviorally constrained neural network modeling, the study sought to distinguish between alternative cognitive strategies. It concludes that humans use a hierarchical information processing strategy, revising earlier decisions using later evidence through counterfactual reasoning.

I really enjoyed reading this paper. The study is well thought out and the writing is very clear. The combination of visual and auditory cues in the H-task is clever and innovative. However, I have some concerns about the conclusion that the data support counterfactual strategies, the interpretation of the fixation patterns, and certain aspects of the neural network model.

1. Counterfactual Reasoning Interpretation. To characterize participants' behavior as "counterfactual" requires that participants make a commitment to a left or right choice after listening to the first interval, and that this initial commitment is sometimes revised after listening to the second interval. Because participants only report the final decision in the task, it is difficult to determine whether an initial commitment occurred. The authors use an analysis of the gaze, which shows that participants often direct their gaze to the left or right bifurcation before the last auditory cue (Supp fig 2 and 3), to support the claim that an initial commitment has occurred. However, there is an alternative interpretation of the gaze data. The proportion of trials in which participants looked at either the left or right bifurcation before the last cue decreases as the difficulty of the initial decision increases (Supplementary Figure 3). This suggests that participants did not make an initial commitment when the initial decision was difficult. Further, even when there was a hierarchical saccade, there is a 78% chance of making a counterfactual saccade (for difficult initial decisions). This suggests that the hierarchical saccade is not really a commitment, as the participant ends up exploring both the left and right bifurcation in most of these trials. These observations invite an alternative model in which participants commit and do not revise their choice when the initial decision is easy ($|evidence| > \text{threshold}$), but remain "uncommitted" when the initial decision is difficult ($|evidence| < \text{threshold}$). In the latter case, participants can rely solely on the second interval to make their choice, for which they will usually need to explore both second-level alternatives. Crucially, this strategy is not "counterfactual" because there is no initial commitment on trials where participants end up considering both second-level alternatives.

Can the author reject such an alternative strategy?

2. Fixation pattern analysis.

The fixation pattern analysis (supplementary figures 2 and 3) is critical and distinguishes this study from others that show that lower-level decisions can bias higher-level decisions (e.g., Lorteije et al., Neuron 2015). I suggest making this a main figure, perhaps combining Supplementary Figures 2 and 3 into a single main figure.

This said, the presentation of the results could be made clearer:

In Supplementary Figure 3, (a) it is unclear why post-feedback fixations (blue traces) are shown, since they do not relate to any of the models; (b) the probabilities shown seem to be conditional probabilities; please indicate what they are being conditioned on; (c) how do these probabilities change when conditioning on choice accuracy (e.g., correct hierarchical saccade, incorrect hierarchical saccade)?

3. Model Comparison.

In comparing models (e.g., Fig. 2d), the authors use the MSE computed between model and participant choice probabilities. Have you considered using the likelihood of the choice (on left-out trials) instead of MSE? Since the models output probabilities, using likelihood should be feasible and offer a more principled approach.

4. Neural network model.

In the model, the auditory inputs 'a1t' and 'a2t' are ramped over time. To make it more similar to the experiment, the inputs a1t and a2t should take a value of 1 when the auditory cue is presented and 0 otherwise. Can the authors explain why a1t and a2t were modeled as ramps?

Minor Points:

- In the legend of Supplementary Figure 2, the text reads: "The pattern of eye movements indicate that participants made their choice hierarchically (early leftward or rightward saccades in the top left panel). They changed their mind occasionally and made a saccade to the other vertical bar either after the second auditory cue (top right), or the third auditory cue (bottom left and right)." It is unclear what is meant by "top left panel", "top right" and "bottom left and right".

- On page 27, the text states: "from switching its attention from side to side every trial." Probably the authors meant "timestep" instead of "trial."

- On page 29, there appears to be an error in the expression "[pLU pLD pRU + pRD]."

- In the last paragraph of the paper (page 32, RNN saccade), the authors refer to a plot in Fig 5b, which lacks a corresponding plot (only a table is presented).

Reviewer #1 (Remarks on code availability):

The code has not been made available for review

Reviewer #2 (Remarks to the Author):

This manuscript "Computational basis of hierarchical and counterfactual information processing" presents a comprehensive study on human decision-making strategies in complex tasks, focusing on the computational constraints that lead to hierarchical processing and counterfactual reasoning. The authors employ an interesting experimental paradigm, the H-maze task, combined with behavioral experiments, eye-tracking, and neural network modeling to investigate these cognitive processes. Overall, this is a strong manuscript that makes a significant contribution to our understanding of human decision-making processes.

Strengths:

1. The study addresses an important question in cognitive science, bridging the gap between cognitive theories and computational models.
2. The experimental design is well-thought-out, with a series of tasks that systematically probe different aspects of decision-making.
3. The combination of human psychophysics and neural network modeling provides converging evidence for the authors' hypotheses.
4. The finding that distinct cognitive strategies may exist on a continuum is novel and thought-provoking.
5. The H-maze task is an interesting methodology that allows for precise quantification of decision strategies.
6. The authors' pre-registration of hypotheses and large-scale online replication enhance the robustness of their findings.
7. The study effectively bridges cognitive psychology, neuroscience, and machine learning, contributing to a more integrated understanding of decision-making.

Weaknesses:

1. The sample size for the in-lab experiments is small (N=16). While this is somewhat mitigated by the online replication (N=150), a discussion of the potential limitations of the small initial sample would be beneficial.
2. The manuscript is quite dense and technical in parts, which may make it challenging for a broader audience to follow. More explanatory figures or simplified technical descriptions could improve accessibility, along with a clear demarcation of what is novel and what is not.

Suggestions for improvement:

1. Structure: Consider reorganizing the manuscript with explicit subheadings for Methodology, Results, Discussion, and

Implications to enhance readability.

2. Prior work comparison: Expand on how this study builds upon or challenges previous research in the field, addressing any conflicting findings or theories.
3. Practical implications: Discuss potential real-world applications of this research, such as in education, AI development, or clinical settings.
4. Limitations: Add a separate section discussing the study's limitations, including those already mentioned and any others that might be relevant.
5. Future directions: Elaborate on potential future research directions that could further validate or extend these findings.
6. Figures: add a more conceptual diagram of what the intent of the manuscript is.

Questions for the authors, please add comments in the manuscript to the extent that word limitations allow:

1. How generalizable are these findings to other types of decision-making tasks? Are there plans for follow-up studies with different paradigms?
2. How well does the RNN model's transition between different strategies depending on noise levels align with individual differences observed in human participants?
3. How might these findings extend to other sensory modalities or more abstract decision-making scenarios?
4. Are there any implications of this work for understanding or treating decision-making deficits in clinical populations?
5. How do the authors' findings relate to other recent work on meta-learning or hierarchical reinforcement learning in AI?

This manuscript presents compelling evidence for the computational constraints underlying hierarchical and counterfactual processing in human decision-making, and it would be a significant contribution to the field.

Reviewer #3 (Remarks to the Author):

This paper investigates the computational basis of hierarchical and counterfactual reasoning in humans. The authors develop a novel decision-making task, called an H-Maze, in which participants must infer the correct path of a ball that becomes hidden from view by matching the timing between auditory clicks to the lengths of the compatible horizontal and vertical arms of the maze. They formalize different cognitive strategies including optimal, hierarchical, postdictive, and counterfactual strategies as computational models and subsequently fit these models to the participants' decisions. This shows that the counterfactual model best explains people's behavior. Then, they use a series of variations of the task to dissect the constraints underlying this strategy. Namely, they show that people have a limited capacity for parallel processing, that there is noise on people's counterfactual processing, and that people utilize counterfactuals in a rational manner. These results all replicate in a large-scale online experiment. Finally, the authors train recurrent neural networks on the task and implemented different combinations of the aforementioned constraints, finding that only the networks with all three constraints behaves similarly to humans.

Overall, my evaluation of the paper is very positive. It sets up an intriguing hypothesis, provides a thorough collection of experiments paired with computational modeling to test that hypothesis, conducts a large-scale replication of the core experiments, and demonstrates a compelling application of neural networks in cognitive science. The paper is also fairly well written. As such, my comments are mainly about exposition and general framing that could be improved.

Major comments

- The introduction frames human counterfactual reasoning as deeply tied to multi-step planning. Specifically, the authors set up a dichotomy between human reliance on counterfactuals when faced with a series of if-then scenarios and the optimal strategies for decision tree search not requiring counterfactual reasoning at all. This is interesting, but I'm not sure the task and subsequent experiments actually address this gap in the literature. The task does seem relatively complex for participants to perform, but it is really a two-step decision-making task. In my view, this is still relatively simple compared to the types of intractable decision trees constructed for complex planning problems that the authors cite. Thus, I would suggest that the framing for the paper be more focused on hierarchical and counterfactual reasoning rather than forcing the connection to multi-step planning. I think that problem is already compelling enough in and of itself that there is no reason to write the introduction otherwise.
- Related to the above, I would like to see much more justification for the task design. Again, the task doesn't strike me as one that primarily requires forward planning with decision trees as is suggested by the introduction. Maybe I'm misunderstanding something, but it seems like much more of a perceptual task in which participants have to map auditory cues onto visual information. If the framing of the paper is changed (or even if it isn't), it would be helpful to see clear commentary about how the task design supports the hypotheses put forth in the introduction.
- I know there is a large-scale replication of the experiments in the paper, but this only comes much later than the initial results are shown. Therefore, the foundational results are presented with a very limited number of participants. Figure 2D in particular highlights this since there is one panel per subject for the model fits. I don't find this very convincing, and even though the large-scale experiments are meant to address this it is still something that I presume readers will be thinking about for a large portion of the paper. Are the laboratory experiments even necessary if you can simply present the large-scale results from the beginning of the paper? I can imagine they might be if the response time and eye movement data made them unique, but analysis of this data is already relegated to the supplement. I would consider making the large-scale data the core of the results if there isn't a compelling reason to avoid doing so.
- I find the discussion surprisingly lacking in how thorough it is compared to the rest of the paper. It's only three paragraphs, and they are all basically a summary of the results. This could be greatly improved with a more standard approach to the discussion that reiterates the findings of the paper in a single paragraph or two, and then elaborates on all of the interesting findings. What other hypotheses can be tested with this framework given the limitations of the current study? What else is there to be learned about counterfactual processing in humans that could be addressed by other experiments? How can the field build off of this work to bridge planning and counterfactual reasoning in complex tasks? There are many points to comment on further here that I would have expected to see.

Minor comments

- Figure 1B is not intuitive to read at all. Is there a simpler way to present the H-Maze arm lengths?
- I had to read the difference between the postdictive and optimal models multiple times in order to understand the scenario in which they would make distinct predictions. An intuitive example here would help, and might actually be useful in the descriptions of each of the models to give a higher-level idea of how each model works.
- Figure 2B: left versus right in the caption should be top versus bottom I think? Also the last sentence mentions animals, which I presume should read participants.
- Figure 2D: "subject" is misspelled in the x-axis of each panel.
- Are there other computational constraints on counterfactual reasoning that have been cited in the literature? How did the authors arrive at these three to test?
- Task variant 1: is there a reason for testing specifically four balls simultaneously? I'm curious if this choice of number is motivated by the literature about parallel processing in any way.
- Figure 5B: I'd be interested in reading more about some of the RNNs that weren't subjected to all of the constraints. For example, the RNN that had the attention and rationality constraints but not counterfactual noise did fit the counterfactual model best, although equally to the optimal model if I'm reading the table correctly. What do the authors suspect is the reason for that? Also, this panel could potentially be better represented as a plot showing the MSEs themselves rather than just reporting the order of the fits.
- There is an entire body of recent literature about using neural networks for cognitive modeling. I think some of these papers should be cited either when the RNN is introduced or in the discussion since this is an emerging field.

Reviewer #3 (Remarks on code availability):

The manuscript states that the code will only be available after peer review, so there was no code to review.

Version 2:

Decision Letter:

24th February 2025

Dear Professor Jazayeri,

Thank you once again for your manuscript, entitled "Computational basis of hierarchical and counterfactual information processing," and for your patience during the peer review process. I apologize for the delay in the peer review process.

Your manuscript has now been evaluated again by 3 reviewers, whose comments are included at the end of this letter. You will see that all 3 reviewers are happy with the revisions made to the manuscript.

However, there is one outstanding issue. The reviewers note that the code underlying this study was not made available. As this is largely a computational study, we believe that it is important that the reviewers have access to the code prior to publication.

We therefore invite you to resubmit your manuscript including a link to the data and code so that the reviewers could reproduce the results. This link should not require any login to access (as this can compromise reviewer confidentiality). OSF.io is one repository which offers no-login links.

I regret the additional delay this will entail. Please do not hesitate to contact us if there are specific requests from the reviewers that you believe are technically impossible or unlikely to yield a meaningful outcome.

We hope to receive your revised manuscript within 1 month. I would be grateful if you could contact us as soon as possible if you foresee difficulties with meeting this target resubmission date.

- Highlight all changes made to your manuscript or provide us with a version that tracks changes.

Link Redacted

We look forward to seeing the revised manuscript and thank you for the opportunity to review your work. Please do not hesitate to contact me if you have any questions or would like to discuss these revisions further.

Sincerely,

██████

██████████████████

██████████████████

Nature Human Behaviour

REVIEWER COMMENTS:

Reviewer #1 (Remarks to the Author):

The authors addressed all my concerns, congratulations for a wonderful study.

Ariel Zylberberg

Reviewer #1 (Remarks on code availability):

No link to the code was provided

Reviewer #2 (Remarks to the Author):

I am satisfied with the reply to my questions and comments, and with the updated manuscript.

Reviewer #3 (Remarks to the Author):

I'd like to thank the authors for their thoughtful responses to my comments. In particular, more precise language about planning and the general framing of the paper in the introduction, focus on the large-scale online data set with complementary analyses, added explanations regarding the RNNs, and new limitations and future directions section of the discussion address the main points I raised and have resulted in a much improved paper. Thus, I am satisfied with the work in its current form.

Reviewer #3 (Remarks on code availability):

The manuscript states that the code will only be available after peer review, so there was no code to review.

Version 3:

Decision Letter:

Our ref: NATHUMBEHAV-24010207C

2nd April 2025

Dear Dr. Jazayeri,

Thank you for submitting your revised manuscript "Computational basis of hierarchical and counterfactual information processing" (NATHUMBEHAV-24010207C). It has now been seen by the original referees and their comments are below. As you can see, the reviewers find that the paper has improved in revision. We will therefore be happy in principle to publish it in Nature Human Behaviour, pending minor revisions to satisfy our editorial and formatting guidelines.

We are now performing detailed checks on your paper and will send you a checklist detailing our editorial and formatting requirements within two weeks. Please do not upload the final materials and make any revisions until you receive this additional information from us.

Sincerely,

██████████

██████████████████

██████████████████

Nature Human Behaviour

Reviewer #1 (Remarks to the Author):

I ran several of the analyses, and the code appears to replicate the results reported in the paper adequately. It is clearly written and easy to execute.

Reviewer #1 (Remarks on code availability):

I ran several of the analyses, and the code appears to replicate the results reported in the paper adequately. It is clearly written and easy to execute.

Reviewer #2 (Remarks to the Author):

I am satisfied with the current version of the manuscript.

Reviewer #2 (Remarks on code availability):

I reviewed the code, and while I didn't run it, it seems consistent and complete. It should be difficult for any researcher to use it: the GitHub repository does include a basic README that instructs how to generate the manuscript figures, so back-tracing from them one can reconstruct the analytic steps.

Reviewer #3 (Remarks on code availability):

The code base associated with this paper appears reasonable. There is a readme file that details where the Matlab and Python scripts and data associated with each figure are located. This decomposition should make it easy for others to replicate findings related to particular figures that may be of interest. The majority of scripts look fairly well commented, although there are some that came across that either didn't have any comments or would benefit from additional comments. I think it would also be helpful if the authors could provide a high-level description of the remaining Python subfolders in the readme in case anyone is interested in the details of, for example, the models or RNNs beyond replicating figures. Otherwise I have no suggestions for improving the code base.

Version 4:

Decision Letter:

Dear Professor Jazayeri,

We are pleased to inform you that your Article "Computational basis of hierarchical and counterfactual information processing", has now been accepted for publication in Nature Human Behaviour.

You can now use a single sign-on for all your accounts, view the status of all your manuscript submissions and reviews, access

usage statistics for your published articles and download a record of your refereeing activity for the Nature journals.

With best regards,

[Redacted]

[Redacted]

Nature Human Behaviour

P.S. Click on the following link if you would like to recommend Nature Human Behaviour to your librarian
<http://www.nature.com/subscriptions/recommend.html#forms>

** Visit the Springer Nature Editorial and Publishing website at http://editorial-jobs.springernature.com?utm_source=ejp_NHumB_email&utm_medium=ejp_NHumB_email&utm_campaign=ejp_NHumB for more information about our career opportunities. If you have any questions please click [here](mailto:editorial.publishing.jobs@springernature.com).

REVIEWER COMMENTS:

Reviewer #1 (Remarks to the Author):

This paper examines how humans approach complex, multi-step decision problems. The authors developed an H-shaped maze task in which participants infer the trajectory of an invisible ball using partial and uncertain cues. By integrating human psychophysics with behaviorally constrained neural network modeling, the study sought to distinguish between alternative cognitive strategies. It concludes that humans use a hierarchical information processing strategy, revising earlier decisions using later evidence through counterfactual reasoning.

I really enjoyed reading this paper. The study is well thought out and the writing is very clear. The combination of visual and auditory cues in the H-task is clever and innovative. However, I have some concerns about the conclusion that the data support counterfactual strategies, the interpretation of the fixation patterns, and certain aspects of the neural network model.

Thank you! It is wonderful to hear that you enjoyed the paper and found it clear and innovative! We also wish to thank you for your thoughtful and constructive comments. We have done our best to address them.

1. Counterfactual Reasoning Interpretation. To characterize participants' behavior as "counterfactual" requires that participants make a commitment to a left or right choice after listening to the first interval, and that this initial commitment is sometimes revised after listening to the second interval. Because participants only report the final decision in the task, it is difficult to determine whether an initial commitment occurred. The authors use an analysis of the gaze, which shows that participants often direct their gaze to the left or right bifurcation before the last auditory cue (Supp fig 2 and 3), to support the claim that an initial commitment has occurred. However, there is an alternative interpretation of the gaze data. The proportion of trials in which participants looked at either the left or right bifurcation before the last cue decreases as the difficulty of the initial decision increases (Supplementary Figure 3). This suggests that participants did not make an initial commitment when the initial decision was difficult. Further, even when there was a hierarchical saccade, there is a 78% chance of making a counterfactual saccade (for difficult initial decisions). This suggests that the hierarchical saccade is not really a commitment, as the participant ends up exploring both the left and right bifurcation in most of these trials. These observations invite an alternative model in which participants commit and do not revise their choice when the initial decision is easy ($|evidence| > \text{threshold}$), but remain "uncommitted" when the initial decision is difficult ($|evidence| < \text{threshold}$). In the latter case, participants can rely solely on the second interval to make their choice, for which they will usually need to explore both second-level alternatives. Crucially, this strategy is not "counterfactual" because there is no initial commitment on trials where participants end up considering both second-level alternatives.

Can the author reject such an alternative strategy?

The reviewer notes the proportion of early eye movements toward left/right drops with the difference between the left and right arm lengths, and suggests that this observation is consistent with an alternative mixed strategy we had not considered: Participants may commit to a left/right decision only when the initial decision is easy but remain uncommitted for difficult initial decisions, relying on the second interval to guide their final choice.

If the proportion of saccades were to drop with difficulty, it would indeed indicate that participants used a mixed strategy. However, in the original submission, we only plotted the proportion of correct saccades – not the total proportion of saccades. The fact that the proportion of correct saccades drops with difficulty is expected and unsurprising. As it turns out, the total proportion of saccades did not show such a pattern.

To address this comment, we have made three revisions. First, we replotted the results including both correct and incorrect. This plot shows that the drop in correct saccades is accompanied by an increase in incorrect saccades with no appreciable drop in the total proportion.

Second, we revised the Discussion highlighting that our work focuses on some of the most prominent latent cognitive models but does not include mixed-strategy models since the space of those models is prohibitively large.

Third, as part of the Discussion, we highlighted one variant of the specific mixed-strategy model noted by the reviewer and added a supplemental figure showing that it is inferior to the counterfactual model.

We are particularly grateful for this comment because it allowed us to highlight a shortcoming of our work regarding mixed-strategy models and our ongoing neurophysiology experiments in monkeys aimed at addressing this question definitively.

2. Fixation pattern analysis.

The fixation pattern analysis (supplementary figures 2 and 3) is critical and distinguishes this study from others that show that lower-level decisions can bias higher-level decisions (e.g., Lorteije et al., Neuron 2015). I suggest making this a main figure, perhaps combining Supplementary Figures 2 and 3 into a single main figure.

This said, the presentation of the results could be made clearer:

In Supplementary Figure 3, (a) it is unclear why post-feedback fixations (blue traces) are shown, since they do not relate to any of the models; (b) the probabilities shown seem to be conditional probabilities; please indicate what they are being conditioned on; (c) how do these probabilities change when conditioning on choice accuracy (e.g., correct hierarchical saccade, incorrect hierarchical saccade)?

Thank you. We have done as suggested. We have removed the post-feedback analysis altogether. We have combined Supplementary Figures 2 and 3 into a single main figure. We

have rewritten the caption to clarify what the probabilities represent and the conditions under which they are calculated. Finally, we have conducted additional analyses investigating how these probabilities change when conditioning on choice accuracy (e.g., correct hierarchical saccades versus incorrect hierarchical saccades).

3. Model Comparison.

In comparing models (e.g., Fig. 2d), the authors use the MSE computed between model and participant choice probabilities. Have you considered using the likelihood of the choice (on left-out trials) instead of MSE? Since the models output probabilities, using likelihood should be feasible and offer a more principled approach.

We did both originally. The results are qualitatively the same. Following this suggestion, we have replaced the MSE analyses with a cross-validated model-based likelihood analysis. For consistency, we also used the same metric for RNNs (Fig. 2d; Fig. 5b; Fig. 6d).

4. Neural network model.

In the model, the auditory inputs 'a1t' and 'a2t' are ramped over time. To make it more similar to the experiment, the inputs a1t and a2t should take a value of 1 when the auditory cue is presented and 0 otherwise. Can the authors explain why a1t and a2t were modeled as ramps?

Over the past decade, we have conducted numerous neurophysiology experiments in monkeys characterizing the neural dynamics associated with measurement and production of time intervals. One of the most important lessons learned from these studies is that encoding of elapsed time can be parsimoniously modeled as a ramping activity with different slopes (Jazayeri & Shadlen, 2015; Wang et al., 2018). The focus of this paper is not on timing mechanisms per se, but rather on the decision-making process. We made the RNN models compatible with what is known about the encoding of elapsed time so we can focus our reverse engineering efforts on the processes that govern decision-making.

To make this motivation more explicit, we have added explanations and citations in the Method section describing the RNN model:

“...For auditory input, although human subjects received discrete auditory clicks that demarcated time intervals, previous studies have shown that time intervals were represented internally as ramping signals between the discrete clicks in the brain and RNNs (Jazayeri & Shadlen, 2015, Wang et al. 2018). To facilitate RNN training, here we directly use the corresponding ramping signal as the time interval input to the RNN...”

Minor Points:

Thank you for reading the manuscript and noticing the following typos and/or errors.

- In the legend of Supplementary Figure 2, the text reads: “The pattern of eye movements indicate that participants made their choice hierarchically (early leftward or rightward saccades in the top left panel). They changed their mind occasionally and made a saccade to the other

vertical bar either after the second auditory cue (top right), or the third auditory cue (bottom left and right).” It is unclear what is meant by “top left panel”, “top right” and “bottom left and right”.

Revised.

- On page 27, the text states: “from switching its attention from side to side every trial.” Probably the authors meant “timestep” instead of “trial.”

Revised.

- On page 29, there appears to be an error in the expression “[pLU pLD pRU + pRD].”

Revised. Since we now use likelihood instead of MSE, this section was removed.

- In the last paragraph of the paper (page 32, RNN saccade), the authors refer to a plot in Fig 5b, which lacks a corresponding plot (only a table is presented).

Revised. Thank you.

Reviewer #2 (Remarks to the Author):

This manuscript "Computational basis of hierarchical and counterfactual information processing" presents a comprehensive study on human decision-making strategies in complex tasks, focusing on the computational constraints that lead to hierarchical processing and counterfactual reasoning. The authors employ an interesting experimental paradigm, the H-maze task, combined with behavioral experiments, eye-tracking, and neural network modeling to investigate these cognitive processes. Overall, this is a strong manuscript that makes a significant contribution to our understanding of human decision-making processes.

Strengths:

1. The study addresses an important question in cognitive science, bridging the gap between cognitive theories and computational models.
2. The experimental design is well-thought-out, with a series of tasks that systematically probe different aspects of decision-making.
3. The combination of human psychophysics and neural network modeling provides converging evidence for the authors' hypotheses.
4. The finding that distinct cognitive strategies may exist on a continuum is novel and thought-provoking.
5. The H-maze task is an interesting methodology that allows for precise quantification of decision strategies.
6. The authors' pre-registration of hypotheses and large-scale online replication enhance the robustness of their findings.
7. The study effectively bridges cognitive psychology, neuroscience, and machine learning, contributing to a more integrated understanding of decision-making.

Thank you! We are grateful for this appraisal! We also wish to thank you for pointing out the following weaknesses, which we have done our best to address.

Weaknesses:

1. The sample size for the in-lab experiments is small (N=16). While this is somewhat mitigated by the online replication (N=150), a discussion of the potential limitations of the small initial sample would be beneficial.

We used the data from the in-lab experiments (N=16) to *generate – not test* – our hypotheses. We used this data to pre-register our hypotheses and analyses and then moved on to testing our hypothesis with the online sample. In other words, we did not treat in-lab participants as part of our sample. However, we understand how we organized the paper may have led to the misunderstanding that we treated that data as part of our experimental sample. To address this point, we have revised the manuscript going directly to presenting the data from the online experiments (N=150).

We now only use in-lab experiments to focus on data that we could not collect online, namely, eye movements and reaction times. We have analyzed the eye-movement data and presented

the key results as a new main figure (Figure 3), following a suggestion by one of the reviewers. This revised analysis provided complementary evidence supporting our hypothesis.

2. The manuscript is quite dense and technical in parts, which may make it challenging for a broader audience to follow. More explanatory figures or simplified technical descriptions could improve accessibility, along with a clear demarcation of what is novel and what is not.

We carefully read the manuscript and made small changes to improve readability. We also asked colleagues to read the paper and highlight any part that could be improved. Finally, we made several changes in response to the three reviewers to help with readability. We would happily make additional revisions if the reviewer highlights other parts that need improvement.

Suggestions for improvement:

1. Structure: Consider reorganizing the manuscript with explicit subheadings for Methodology, Results, Discussion, and Implications to enhance readability.

Thank you for the comments. We are unsure what the reviewer has in mind since the current manuscript has several subheadings along the lines suggested.

2. Prior work comparison: Expand on how this study builds upon or challenges previous research in the field, addressing any conflicting findings or theories.

We have revised the Introduction to highlight the prior work and the gaps in our understanding. Specifically, we explain how our work focuses on the interplay between hierarchical and counterfactual reasoning, and identifies the computational constraints that lead to these behavioral strategies. We have also revised the Discussion adding a section on Limitations and Future Directions highlighting how our work builds in previous studies and what key questions remain unanswered.

3. Practical implications: Discuss potential real-world applications of this research, such as in education, AI development, or clinical settings.

Our work addressed fundamental basic science questions with numerous potential applications. To avoid the risk of overselling our work, we have highlighted only a few general application domains in AI and health in the new Limitations and Future Directions section of the Discussion.

4. Limitations: Add a separate section discussing the study's limitations, including those already mentioned and any others that might be relevant. The new Limitations and Future Directions of the Discussion addresses this issue.

Thank you for this suggestion. We had not done this properly. We hope the new Limitations and Future Directions section of the Discussion addresses this shortcoming.

5. Future directions: Elaborate on potential future research directions that could further validate or extend these findings.

This is included in the same newly added section of the Discussion

6. Figures: add a more conceptual diagram of what the intent of the manuscript is.

We used conceptual figures in places we could think of (e.g., tasks and algorithms). We are unsure where else we can help with this but we will do so if the reviewer has specific additional suggestions.

Questions for the authors, please add comments in the manuscript to the extent that word limitations allow:

1. How generalizable are these findings to other types of decision-making tasks? Are there plans for follow-up studies with different paradigms?

As part of the newly added Future Directions, we now highlight various ways our findings may be extended across other decision-making paradigms. We have also highlighted that our immediate next step is to conduct these experiments in nonhuman primates with neurophysiology to scrutinize our findings with neural data.

2. How well does the RNN model's transition between different strategies depending on noise levels align with individual differences observed in human participants?

This was precisely our rationale for designing and running Task Variant 3. In this variant, we probed the relationship between counterfactual probability and memory noise (inferred from Task Variant 2). Our findings reveal that participants' reliance on counterfactuals changes lawfully with noise. This is aligned with the behavior of RNNs with various levels of noise.

3. How might these findings extend to other sensory modalities or more abstract decision-making scenarios?

We do not know but have noted this in the revised Discussion.

4. Are there any implications of this work for understanding or treating decision-making deficits in clinical populations?

This is related to the reviewer's previous comments (suggestion 3).

5. How do the authors' findings relate to other recent work on meta-learning or hierarchical reinforcement learning in AI?

We do not know but have noted this in the revised Discussion (see response to suggestion 3).

This manuscript presents compelling evidence for the computational constraints underlying hierarchical and counterfactual processing in human decision-making, and it would be a significant contribution to the field.

Thank you! We are grateful for your thoughtful comments.

Reviewer #3 (Remarks to the Author):

This paper investigates the computational basis of hierarchical and counterfactual reasoning in humans. The authors develop a novel decision-making task, called an H-Maze, in which participants must infer the correct path of a ball that becomes hidden from view by matching the timing between auditory clicks to the lengths of the compatible horizontal and vertical arms of the maze. They formalize different cognitive strategies including optimal, hierarchical, postdictive, and counterfactual strategies as computational models and subsequently fit these models to the participants' decisions. This shows that the counterfactual model best explains people's behavior. Then, they use a series of variations of the task to dissect the constraints underlying this strategy. Namely, they show that people have a limited capacity for parallel processing, that there is noise on people's counterfactual processing, and that people utilize counterfactuals in a rational manner. These results all replicate in a large-scale online experiment. Finally, the authors train recurrent neural networks on the task and implemented different combinations of the aforementioned constraints, finding that only the networks with all three constraints behaves similarly to humans.

Overall, my evaluation of the paper is very positive. It sets up an intriguing hypothesis, provides a thorough collection of experiments paired with computational modeling to test that hypothesis, conducts a large-scale replication of the core experiments, and demonstrates a compelling application of neural networks in cognitive science. The paper is also fairly well written. As such, my comments are mainly about exposition and general framing that could be improved.

Thank you! We are grateful for the positive feedback!

Major comments

- The introduction frames human counterfactual reasoning as deeply tied to multi-step planning. Specifically, the authors set up a dichotomy between human reliance on counterfactuals when faced with a series of if-then scenarios and the optimal strategies for decision tree search not requiring counterfactual reasoning at all. This is interesting, but I'm not sure the task and subsequent experiments actually address this gap in the literature. The task does seem relatively complex for participants to perform, but it is really a two-step decision-making task. In my view, this is still relatively simple compared to the types of intractable decision trees constructed for complex planning problems that the authors cite. Thus, I would suggest that the framing for the paper be more focused on hierarchical and counterfactual reasoning rather than forcing the connection to multi-step planning. I think that problem is already compelling enough in and of itself that there is no reason to write the introduction otherwise.

We fully agree with this assessment. We had discussed internally whether and how our work might relate to planning and decided that the connection, if any, is too tenuous to highlight. Accordingly, we deliberately avoided the word "planning" throughout the Introduction and Results. We regret that, despite our best intentions, the Introduction gave the impression that it addresses the link between counterfactual reasoning and multi-step planning. We revised the Introduction to draw the focus on inference in decision trees, and not planning. We also have

expanded the Discussion stating explicitly that future work is needed to study the link between planning and counterfactual reasoning. We hope these revisions address the reviewer's comment.

- Related to the above, I would like to see much more justification for the task design. Again, the task doesn't strike me as one that primarily requires forward planning with decision trees as is suggested by the introduction. Maybe I'm misunderstanding something, but it seems like much more of a perceptual task in which participants have to map auditory cues onto visual information. If the framing of the paper is changed (or even if it isn't), it would be helpful to see clear commentary about how the task design supports the hypotheses put forth in the introduction.

Thank you for this comment. We have revised the Introduction to address this comment. Specifically, after reviewing prior literature, we highlight the open questions our work aims to address:

“However, several important questions about the computational basis of counterfactual reasoning have remained unanswered. What computational constraints motivate reliance on counterfactuals? Do humans adopt a computationally rational approach to using counterfactuals? Does computing counterfactuals introduce sub-optimality in behavior? If so, what are they?”

We then describe our task design highlighting the key features that make it suitable for addressing these questions:

“We developed a simple and intuitive decision-making task to answer these questions. The task required participants to infer the path of an invisible moving ball within a maze using partial and uncertain cues (Fig. 1a). The task featured two key desiderata needed to investigate the computational characteristics of hierarchical decision-making and counterfactual information processing. First, the maze geometry confronted participants with a two-stage hierarchically organized decision-making problem. Second, parametric control of uncertainty at each decision stage afforded parametric control over the degree to which counterfactual revisions could improve decisions.”

- I know there is a large-scale replication of the experiments in the paper, but this only comes much later than the initial results are shown. Therefore, the foundational results are presented with a very limited number of participants. Figure 2D in particular highlights this since there is one panel per subject for the model fits. I don't find this very convincing, and even though the large-scale experiments are meant to address this it is still something that I presume readers will be thinking about for a large portion of the paper. Are the laboratory experiments even necessary if you can simply present the large-scale results from the beginning of the paper? I can imagine they might be if the response time and eye movement data made them unique, but analysis of this data is already relegated to the supplement. I would consider making the large-scale data the core of the results if there isn't a compelling reason to avoid doing so.

Thank you for this suggestion. We have restructured the manuscript with the main figures focusing on the large-scale online data.

The in-lab experiments provide unique data related to eye movements and reaction time. As suggested by other reviewers, we now use the in-lab data to present complementary results related to eye movements (revised main figure 3) and supplementary results related to reaction time data.

- I find the discussion surprisingly lacking in how thorough it is compared to the rest of the paper. It's only three paragraphs, and they are all basically a summary of the results. This could be greatly improved with a more standard approach to the discussion that reiterates the findings of the paper in a single paragraph or two, and then elaborates on all of the interesting findings. What other hypotheses can be tested with this framework given the limitations of the current study? What else is there to be learned about counterfactual processing in humans that could be addressed by other experiments? How can the field build off of this work to bridge planning and counterfactual reasoning in complex tasks? There are many points to comment on further here that I would have expected to see.

We fully agree and regret not having done this in the first place. We have expanded the Discussion in various ways. Most importantly, we have added an extensive "Limitations and Future Directions" section. In this section, we discuss the implications and shortcomings of our work, and future questions and experiments that can build on our work. We also highlight the need for more complex tasks to bridge the gap in understanding the link between planning and counterfactual reasoning.

Minor comments

- Figure 1B is not intuitive to read at all. Is there a simpler way to present the H-Maze arm lengths?

Great suggestion. We replaced this panel with a more intuitive and simplified demonstration, which we hope addresses this comment.

- I had to read the difference between the postdictive and optimal models multiple times in order to understand the scenario in which they would make distinct predictions. An intuitive example here would help, and might actually be useful in the descriptions of each of the models to give a higher-level idea of how each model works.

We agree. The difference between the postdictive and optimal models is not as intuitive as the difference between the other models. We have revised the text to clarify (see below). We would appreciate the reviewer commenting on whether this is sufficient:

"Note that the postdictive model differs from the optimal model in how the likelihoods of the four vertical arms enter the calculations. The optimal model treats the likelihoods of the four vertical arms separately. The postdictive model, in contrast, relies on the sum of the vertical likelihoods for each side to make its left/right decision. This summation makes the postdictive strategy

suboptimal because the sum is blind to the individual likelihoods. For example, imagine a trial with a very high and very low vertical likelihood on the left, and two intermediate likelihoods on the right. The optimal model will take full advantage of these differences. The postdictive model, in comparison, will have less discriminative power because the two sums will be similar.”

- Figure 2B: left versus right in the caption should be top versus bottom I think? Also the last sentence mentions animals, which I presume should read participants.

Revised.

- Figure 2D: "subject" is misspelled in the x-axis of each panel.

This subpanel has now changed following reviewers' suggestions.

- Are there other computational constraints on counterfactual reasoning that have been cited in the literature? How did the authors arrive at these three to test?

To our knowledge, the computational constraints that motivate counterfactual reasoning are not systematically studied. However, it seems plausible that other constraints may also be at play. We have added a new "Limitations and Future Directions" section in the Discussion elaborating on how we selected the three constraints tested in this study and discussing other computational constraints.

- Task variant 1: is there a reason for testing specifically four balls simultaneously? I'm curious if this choice of number is motivated by the literature about parallel processing in any way.

We have revised the text describing how the design of Task Variant 1 mirrors the computational challenge of the original H-maze task; i.e., simultaneously updating four posterior probabilities. We hope this clarification helps contextualize the choice within the framework of the task.

- Figure 5B: I'd be interested in reading more about some of the RNNs that weren't subjected to all of the constraints. For example, the RNN that had the attention and rationality constraints but not counterfactual noise did fit the counterfactual model best, although equally to the optimal model if I'm reading the table correctly. What do the authors suspect is the reason for that?

We agree that some RNN variants are interesting in their own right. We have included a section in the Main text to discuss all the RNN variants:

“We evaluated the response patterns of all RNN variants by computing the log-likelihood of the RNN's choices relative to those associated with different cognitive strategies (Fig. 5b, see Methods). RNNs without the attention bottleneck produced response patterns associated with the optimal strategy (Fig. 5b, columns 1 and 2). This result is expected because RNNs without the attention bottleneck can simultaneously evaluate all possible exits. Incorporating the attention bottleneck caused the RNN to produce response patterns associated with the

hierarchical strategy (Fig. 5b, columns 3 and 4). The addition of the rationality constraint enabled the RNN to shift its attention between the two sides and generate responses similar to the optimal model (Fig. 5b, column 5). Finally, adding counterfactual processing noise enabled the RNN to balance the benefit of computing counterfactuals with the degrading effect of the counterfactual noise. The resulting model produced responses that combined the hierarchical and counterfactual strategies and best matched our participants' responses (Fig. 5b, column 6). We refer to this RNN, which was subjected to all constraints, as RNN_{best} ."

Also, this panel could potentially be better represented as a plot showing the MSEs themselves rather than just reporting the order of the fits.

Agreed. We have revised the figure showing the comparisons quantitatively (updated Figure 5b). Please note that in response to another reviewer's comments, we have replaced the MSE with negative log-likelihood, which is a more principled metric for model comparison.

- There is an entire body of recent literature about using neural networks for cognitive modeling. I think some of these papers should be cited either when the RNN is introduced or in the discussion since this is an emerging field.

Agreed. In the revised manuscript, we have included additional citations highlighting relevant work that used neural network models to understand cognition:

"However, human experiments alone are not sufficient for evaluating the causal link between these characteristics and the participants' strategies in the H-maze task. To address this shortcoming, we used neural network models, which offer a powerful platform for testing which computational constraints are critical for emulating human-like cognitive strategies (Rafiei et al. 2024; Sporer et al. 2020; Tsvetkov et al. 2023; Lake and Baroni 2023). Accordingly, we developed multiple task-optimized recurrent neural networks (RNN) and subjected them to different combinations of these constraints to test which ones would generate behavioral response patterns compatible with humans."

"However, human experiments alone can hardly demonstrate the causal relationship between fundamental cognitive architecture and high-level behavioral strategy because we cannot experimentally manipulate subjects' cognitive parameters. Neural network modeling has recently become a powerful tool to bridge the gap. One can parametrically perturb the architecture, learning rule, objective function, regularizations, and so on in models to identify the core factors that lead to human-like cognition (Lake & Baroni, 2023; Rafiei et al., 2024; Sporer et al., 2020; Tsvetkov et al., 2023)."